# A Unified Approach to Domain Incremental Learning with Memory: Theory and Algorithm

**Haizhou Shi**
Department of Computer Science
Rutgers University
Piscataway, NJ 08854
haizhou.shi@rutgers.edu

**Hao Wang**
Department of Computer Science
Rutgers University
Piscataway, NJ 08854
hw488@cs.rutgers.edu

## Abstract

Domain incremental learning aims to adapt to a sequence of domains with access to only a small subset of data (i.e., memory) from previous domains. Various methods have been proposed for this problem, but it is still unclear how they are related and when practitioners should choose one method over another. In response, we propose a unified framework, dubbed Unified Domain Incremental Learning (UDIL), for domain incremental learning with memory. Our UDIL unifies various existing methods, and our theoretical analysis shows that UDIL always achieves a tighter generalization error bound compared to these methods. The key insight is that different existing methods correspond to our bound with different *fixed* coefficients; based on insights from this unification, our UDIL allows *adaptive* coefficients during training, thereby always achieving the tightest bound. Empirical results show that our UDIL outperforms the state-of-the-art domain incremental learning methods on both synthetic and real-world datasets. Code will be available at https://github.com/Wang-ML-Lab/unified-continual-learning.

## 1 Introduction

Despite recent success of large-scale machine learning models [35, 48, 36, 28, 92, 22, 33], continually learning from evolving environments remains a longstanding challenge. Unlike the conventional machine learning paradigms where learning is performed on a static dataset, *domain incremental learning, i.e., continual learning with evolving domains*, hopes to accommodate the model to the dynamically changing data distributions, while retaining the knowledge learned from previous domains [90, 60, 41, 97, 27]. Naive methods, such as continually finetuning the model on new-coming domains, will suffer a substantial performance drop on the previous domains; this is referred to as "catastrophic forgetting" [46, 58, 81, 105, 52]. In general, domain incremental learning algorithms aim to minimize the total risk of *all* domains, i.e.,

$$\mathcal{L}^*(\theta) = \mathcal{L}_t(\theta) + \mathcal{L}_{1:t-1}(\theta) = \mathbb{E}_{(x,y)\sim\mathcal{D}_t}[\ell(y, h_\theta(x))] + \sum_{i=1}^{t-1}\mathbb{E}_{(x,y)\sim\mathcal{D}_i}[\ell(y, h_\theta(x))], \quad (1)$$

where $\mathcal{L}_t$ calculates model $h_\theta$'s expected prediction error $\ell$ over the current domain's data distribution $\mathcal{D}_t$. $\mathcal{L}_{1:t-1}$ is the total error evaluated on the past $t-1$ domains' data distributions, i.e., $\{\mathcal{D}_i\}_{i=1}^{t-1}$.

The main challenge of domain incremental learning comes from the practical *memory constraint* that no (or only very limited) access to the past domains' data is allowed [52, 46, 105, 74]. Under such a constraint, it is difficult, if not impossible, to accurately estimate and optimize the past error $\mathcal{L}_{1:t-1}$. Therefore the main focus of recent domain incremental learning methods has been to develop effective surrogate learning objectives for $\mathcal{L}_{1:t-1}$. Among these methods [46, 81, 2, 105, 58, 10, 75,

77, 21, 25, 65, 66, 9, 72, 82, 95, 53], replay-based methods, which replay a small set of old exemplars during training [90, 75, 8, 4, 80, 11], has consistently shown promise and is therefore commonly used in practice.

One typical example is ER [75], which stores a set of exemplars $\mathcal{M}$ and uses a replay loss $\mathcal{L}_{\text{replay}}$ as the surrogate of $\mathcal{L}_{1:t-1}$. In addition, a fixed, predetermined coefficient $\beta$ is used to balance current domain learning and past sample replay. Specifically,

$$\widetilde{\mathcal{L}}(\theta) = \mathcal{L}_t(\theta) + \beta \cdot \mathcal{L}_{\text{replay}}(\theta) = \mathcal{L}_t(\theta) + \beta \cdot \mathbb{E}_{(x',y')\sim\mathcal{M}}[\ell(y', h_\theta(x'))]. \tag{2}$$

While such methods are popular in practice, there is still a gap between the surrogate loss ($\beta\mathcal{L}_{\text{replay}}$) and the true objective ($\mathcal{L}_{1:t-1}$), rendering them lacking in theoretical support and therefore calling into question their reliability. Besides, different methods use different schemes of setting $\beta$ [75, 8, 4, 80], and it is unclear how they are related and when practitioners should choose one method over another.

To address these challenges, we develop a unified generalization error bound and theoretically show that different existing methods are actually minimizing the same error bound with different *fixed* coefficients (more details in Table 1 later). Based on such insights, we then develop an algorithm that allows *adaptive* coefficients during training, thereby always achieving the tightest bound and improving the performance. Our contributions are as follows:

- We propose a unified framework, dubbed Unified Domain Incremental Learning (UDIL), for domain incremental learning with memory to unify various existing methods.
- Our theoretical analysis shows that different existing methods are equivalent to minimizing the same error bound with different *fixed* coefficients. Based on insights from this unification, our UDIL allows *adaptive* coefficients during training, thereby always achieving the tightest bound and improving the performance.
- Empirical results show that our UDIL outperforms the state-of-the-art domain incremental learning methods on both synthetic and real-world datasets.

## 2 Related Work

**Continual Learning.** Prior work on continual learning can be roughly categorized into three scenarios [90, 15]: (i) task-incremental learning, where task indices are available during both training and testing [52, 46, 90], (ii) class-incremental learning, where new classes are incrementally included for the classifier [74, 100, 30, 45, 44], and (iii) domain-incremental learning, where the data distribution's incremental shift is explicitly modeled [60, 41, 97, 27]. Regardless of scenarios, the main challenge of continual learning is to alleviate catastrophic forgetting with only limited access to the previous data; therefore methods in one scenario can often be easily adapted for another. Many methods have been proposed to tackle this challenge, including functional and parameter regularization [52, 46, 81, 2], constraining the optimization process [77, 21, 58, 10], developing incrementally updated components [104, 38, 53], designing modularized model architectures [73, 95], improving representation learning with additional inductive biases [9, 66, 65, 25], and Bayesian approaches [24, 63, 49, 1]. Among them, replaying a small set of old exemplars, i.e., memory, during training has shown great promise as it is easy to deploy, *applicable in all three scenarios*, and, most importantly, achieves impressive performance [90, 75, 8, 4, 80, 11]. Therefore in this paper, we focus on *domain incremental learning* with *memory*, aiming to provide a principled theoretical framework to unify these existing methods.

**Domain Adaptation and Domain Incremental Learning.** Loosely related to our work are domain adaptation (DA) methods, which adapt a model trained on *labeled* source domains to *unlabeled* target domains [68, 67, 57, 78, 79, 108, 71, 16, 17, 64, 94, 51]. Much prior work on DA focuses on matching the distribution of the source and target domains by directly matching the statistical attributions [67, 89, 87, 71, 64] or adversarial training [108, 57, 26, 109, 17, 102, 101, 54, 94]. Compared to DA's popularity, domain incremental learning (DIL) has received limited attention in the past. However, it is now gaining significant traction in the research community [90, 60, 41, 97, 27]. These studies predominantly focus on the practical applications of DIL, such as semantic segmentation [27], object detection for autonomous driving [60], and learning continually in an open-world setting [18]. Inspired by the theoretical foundation of adversarial DA [5, 57], we propose, to the best of our knowledge, **the first unified upper bound for DIL**. Most related to our work are previous DA methods that flexibly align different domains according to their associated given

or inferred domain index [94, 101], domain graph [102], and domain taxonomy [54]. The main difference between DA and DIL is that the former focuses on improving the accuracy of the *target domains*, while the latter focuses on the total error of *all domains*, with additional measures taken to alleviate forgetting on the previous domains. More importantly, DA methods typically require access to target domain data to match the distributions, and therefore are not directly applicable to DIL.

# 3 Theory: Unifying Domain Incremental Learning

In this section, we formalize the problem of domain incremental learning, provide the generalization bound of naively applying empirical risk minimization (ERM) on the memory bank, derive two error bounds (i.e., intra-domain and cross-domain error bounds) more suited for domain incremental learning, and then unify these three bounds to provide our final adaptive error bound. We then develop an algorithm inspired by this bound in Sec. 4. **All proofs of lemmas, theorems, and corollaries can be found in Appendix A.**

**Problem Setting and Notation.** We consider the problem of domain incremental learning with $T$ domains arriving one by one. Each domain $i$ contains $N_i$ data points $\mathcal{S}_i = \{(\boldsymbol{x}_j^{(i)}, y_j^{(i)})\}_{j=1}^{N_i}$, where $(\boldsymbol{x}_j^{(i)}, y_j^{(i)})$ is sampled from domain $i$'s data distribution $\mathcal{D}_i$. Assume that when domain $t \in [T] \triangleq \{1, 2, \ldots, T\}$ arrives at time $t$, one has access to (1) the current domain $t$'s data $\mathcal{S}_t$, (2) a memory bank $\mathcal{M} = \{M_i\}_{i=1}^{t-1}$, where $M_i = \{(\widetilde{\boldsymbol{x}}_j^{(i)}, \widetilde{y}_j^{(i)})\}_{j=1}^{\widetilde{N}_i}$ is a small subset ($\widetilde{N}_i \ll N_i$) randomly sampled from $\mathcal{S}_i$, and (3) the history model $H_{t-1}$ after training on the previous $t - 1$ domains. For convenience we use shorthand notation $\mathcal{X}_i \triangleq \{\boldsymbol{x}_j^{(i)}\}_{j=1}^{N_i}$ and $\widetilde{\mathcal{X}}_i \triangleq \{\widetilde{\boldsymbol{x}}_j^{(i)}\}_{j=1}^{\widetilde{N}_i}$. The goal is to learn the optimal model (hypothesis) $h^*$ that minimizes the prediction error over all $t$ domains after each domain $t$ arrives. Formally,

$$h^* = \arg\min_h \sum_{i=1}^t \epsilon_{\mathcal{D}_i}(h), \qquad \epsilon_{\mathcal{D}_i}(h) \triangleq \mathbb{E}_{\boldsymbol{x} \sim \mathcal{D}_i}[h(\boldsymbol{x}) \neq f_i(\boldsymbol{x})], \qquad (3)$$

where for domain $i$, we assume the labels $y \in \mathcal{Y} = \{0, 1\}$ are produced by an unknown deterministic function $y = f_i(\boldsymbol{x})$ and $\epsilon_{\mathcal{D}_i}(h)$ denotes the expected error of domain $i$.

## 3.1 Naive Generalization Bound Based on ERM

**Definition 3.1** (**Domain-Specific Empirical Risks**). *When domain $t$ arrives, model $h$'s empirical risk $\widehat{\epsilon}_{\mathcal{D}_i}(h)$ for each domain $i$ (where $i \leq t$) is computed on the available data at time $t$, i.e.,*

$$\widehat{\epsilon}_{\mathcal{D}_i}(h) = \begin{cases} \frac{1}{N_i} \sum_{\boldsymbol{x} \in X_i} \mathbb{1}_{h(\boldsymbol{x}) \neq f_i(\boldsymbol{x})} & \text{if } i = t, \\ \frac{1}{\widetilde{N}_i} \sum_{\boldsymbol{x} \in \widetilde{X}_i} \mathbb{1}_{h(\boldsymbol{x}) \neq f_i(\boldsymbol{x})} & \text{if } i < t. \end{cases} \qquad (4)$$

Note that at time $t$, only a small subset of data from previous domains ($i < t$) is available in the memory bank ($\widetilde{N}_i \ll N_i$). Therefore empirical risks for previous domains ($\widehat{\epsilon}_{\mathcal{D}_i}(h)$ with $i < t$) can deviate a lot from the true risk $\epsilon_{\mathcal{D}_i}(h)$ (defined in Eqn. 3); this is reflected in Lemma 3.1 below.

**Lemma 3.1** (**ERM-Based Generalization Bound**). *Let $\mathcal{H}$ be a hypothesis space of VC dimension $d$. When domain $t$ arrives, there are $N_t$ data points from domain $t$ and $\widetilde{N}_i$ data points from each previous domain $i < t$ in the memory bank. With probability at least $1 - \delta$, we have:*

$$\sum_{i=1}^t \epsilon_{\mathcal{D}_i}(h) \leq \sum_{i=1}^t \widehat{\epsilon}_{\mathcal{D}_i}(h) + \sqrt{\left(\frac{1}{N_t} + \sum_{i=1}^{t-1} \frac{1}{\widetilde{N}_i}\right)\left(8d \log\left(\frac{2eN}{d}\right) + 8 \log\left(\frac{2}{\delta}\right)\right)}. \qquad (5)$$

Lemma 3.1 shows that naively using ERM to learn $h$ is equivalent to minimizing a loose generalization bound in Eqn. 33. Since $\widetilde{N}_i \ll N_i$, there is a large constant $\sum_{i=1}^{t-1} \frac{1}{\widetilde{N}_i}$ compared to $\frac{1}{N_t}$, making the second term of Eqn. 33 much larger and leading to a looser bound.

## 3.2 Intra-Domain and Cross-Domain Model-Based Bounds

In domain incremental learning, one has access to the history model $H_{t-1}$ besides the memory bank $\{M_i\}_{i=1}^{t-1}$; this offers an opportunity to derive tighter error bounds, potentially leading to better algorithms. In this section, we will derive two such bounds, an intra-domain error bound (Lemma 3.2) and a cross-domain error bound (Lemma 3.3), and then integrate them two with the ERM-based bound in Eqn. 33 to arrive at our final adaptive bound (Theorem 3.4).

**Lemma 3.2 (Intra-Domain Model-Based Bound).** *Let $h \in \mathcal{H}$ be an arbitrary function in the hypothesis space $\mathcal{H}$, and $H_{t-1}$ be the model trained after domain $t-1$. The domain-specific error $\epsilon_{\mathcal{D}_i}(h)$ on the previous domain $i$ has an upper bound:*

$$\epsilon_{\mathcal{D}_i}(h) \leq \epsilon_{\mathcal{D}_i}(h, H_{t-1}) + \epsilon_{\mathcal{D}_i}(H_{t-1}), \tag{6}$$

*where $\epsilon_{\mathcal{D}_i}(h, H_{t-1}) \triangleq \mathbb{E}_{\boldsymbol{x} \sim \mathcal{D}_i}[h(\boldsymbol{x}) \neq H_{t-1}(\boldsymbol{x})]$.*

Lemma 3.2 shows that the current model $h$'s error on domain $i$ is bounded by the discrepancy between $h$ and the history model $H_{t-1}$ plus the error of $H_{t-1}$ on domain $i$.

One potential issue with the bound Eqn. 34 is that only a limited number of data is available for each previous domain $i$ in the memory bank, making empirical estimation of $\epsilon_{\mathcal{D}_i}(h, H_{t-1}) + \epsilon_{\mathcal{D}_i}(H_{t-1})$ challenging. Lemma 3.3 therefore provides an alternative bound.

**Lemma 3.3 (Cross-Domain Model-Based Bound).** *Let $h \in \mathcal{H}$ be an arbitrary function in the hypothesis space $\mathcal{H}$, and $H_{t-1}$ be the function trained after domain $t-1$. The domain-specific error $\epsilon_{\mathcal{D}_i}(h)$ evaluated on the previous domain $i$ then has an upper bound:*

$$\epsilon_{\mathcal{D}_i}(h) \leq \epsilon_{\mathcal{D}_t}(h, H_{t-1}) + \tfrac{1}{2}d_{\mathcal{H}\Delta\mathcal{H}}(\mathcal{D}_i, \mathcal{D}_t) + \epsilon_{\mathcal{D}_i}(H_{t-1}), \tag{7}$$

*where $d_{\mathcal{H}\Delta\mathcal{H}}(\mathcal{P}, \mathcal{Q}) = 2\sup_{h \in \mathcal{H}\Delta\mathcal{H}} |\Pr_{x \sim \mathcal{P}}[h(x) = 1] - \Pr_{x \sim \mathcal{Q}}[h(x) = 1]|$ denotes the $\mathcal{H}\Delta\mathcal{H}$-divergence between distribution $\mathcal{P}$ and $\mathcal{Q}$, and $\epsilon_{\mathcal{D}_t}(h, H_{t-1}) \triangleq \mathbb{E}_{\boldsymbol{x} \sim \mathcal{D}_t}[h(\boldsymbol{x}) \neq H_{t-1}(\boldsymbol{x})]$.*

Lemma 3.3 shows that if the divergence between domain $i$ and domain $t$, i.e., $d_{\mathcal{H}\Delta\mathcal{H}}(\mathcal{D}_i, \mathcal{D}_t)$, is small enough, one can use $H_{t-1}$'s predictions evaluated on the current domain $\mathcal{D}_t$ as a surrogate loss to prevent catastrophic forgetting. Compared to the error bound Eqn. 34 which is hindered by limited data from previous domains, Eqn. 35 relies on the current domain $t$ which contains abundant data and therefore enjoys much lower generalization error. Our lemma also justifies LwF-like cross-domain distillation loss $\epsilon_{\mathcal{D}_t}(h, H_{t-1})$ which are widely adopted [52, 23, 100].

## 3.3 A Unified and Adaptive Generalization Error Bound

Our Lemma 3.1, Lemma 3.2, and Lemma 3.3 provide three different ways to bound the true risk $\sum_{i=1}^{t} \epsilon_{\mathcal{D}_i}(h)$; each has its own advantages and disadvantages. Lemma 3.1 overly relies on the limited number of data points from previous domains $i < t$ in the memory bank to compute the empirical risk; Lemma 3.2 leverages the history model $H_{t-1}$ for knowledge distillation, but is still hindered by the limited number of data points in the memory bank; Lemma 3.3 improves over Lemma 3.2 by leveraging the abundant data $\mathcal{D}_t$ in the current domain $t$, but only works well if the divergence between domain $i$ and domain $t$, i.e., $d_{\mathcal{H}\Delta\mathcal{H}}(\mathcal{D}_i, \mathcal{D}_t)$, is small. Therefore, we propose to integrate these three bounds using coefficients $\{\alpha_i, \beta_i, \gamma_i\}_{i=1}^{t-1}$ (with $\alpha_i + \beta_i + \gamma_i = 1$) in the theorem below.

**Theorem 3.4 (Unified Generalization Bound for All Domains).** *Let $\mathcal{H}$ be a hypothesis space of VC dimension $d$. Let $N = N_t + \sum_i^{t-1} \widetilde{N}_i$ denoting the total number of data points available to the training of current domain $t$, where $N_t$ and $\widetilde{N}_i$ denote the numbers of data points collected at domain $t$ and data points from the previous domain $i$ in the memory bank, respectively. With probability at*

Table 1: **UDIL as a unified framework** for domain incremental learning with memory. Three methods (LwF [52], ER [75], and DER++ [8]) are by default compatible with DIL setting. For the remaining four CIL methods (iCaRL [74], CLS-ER [4], EMS-ER [80], and BiC [100]), we adapt their original training objective to DIL settings before the analysis. For CLS-ER [4] and EMS-ER [80], $\lambda$ and $\lambda'$ are the intensity coefficients of the logits distillation. For BiC [100], $t$ is the current number of the incremental domain. The conditions under which the unification of each method is achieved are provided in detail in Appendix B.

|  | UDIL (Ours) | LwF [52] | ER [75] | DER++ [8] | iCaRL [74] | CLS-ER [4] | EMS-ER [80] | BiC [100] |
|---|---|---|---|---|---|---|---|---|
| $\alpha_i$ | $[0,1]$ | 0 | 0 | 0.5 | 1 | $\lambda/(1+\lambda)$ | $\lambda'/(1+\lambda')$ | $1/(2t-1)$ |
| $\beta_i$ | $[0,1]$ | 1 | 0 | 0 | 0 | 0 | 0 | $(t-1)/(2t-1)$ |
| $\gamma_i$ | $[0,1]$ | 0 | 1 | 0.5 | 0 | $1/(1+\lambda)$ | $1/(1+\lambda')$ | $t-1/(2t-1)$ |

*least $1 - \delta$, we have:*

$$
\sum_{i=1}^{t} \epsilon_{\mathcal{D}_i}(h) \leq \left\{ \sum_{i=1}^{t-1} [\gamma_i \widehat{\epsilon}_{\mathcal{D}_i}(h) + \alpha_i \widehat{\epsilon}_{\mathcal{D}_i}(h, H_{t-1})] \right\} + \left\{ \widehat{\epsilon}_{\mathcal{D}_t}(h) + (\sum_{i=1}^{t-1} \beta_i) \widehat{\epsilon}_{\mathcal{D}_t}(h, H_{t-1}) \right\}
$$

$$
+ \tfrac{1}{2} \sum_{i=1}^{t-1} \beta_i d_{\mathcal{H}\Delta\mathcal{H}}(\mathcal{D}_i, \mathcal{D}_t) + \sum_{i=1}^{t-1} (\alpha_i + \beta_i) \epsilon_{\mathcal{D}_i}(H_{t-1})
$$

$$
+ \sqrt{\left( \frac{(1+\sum_{i=1}^{t-1} \beta_i)^2}{N_t} + \sum_{i=1}^{t-1} \frac{(\gamma_i + \alpha_i)^2}{\widetilde{N}_i} \right) \left( 8d \log\left(\frac{2eN}{d}\right) + 8 \log\left(\frac{2}{\delta}\right) \right)}
$$

$$
\triangleq g(h, H_{t-1}, \Omega), \tag{8}
$$

*where $\widehat{\epsilon}_{\mathcal{D}_i}(h, H_{t-1}) = \frac{1}{\widetilde{N}_i} \sum_{\boldsymbol{x} \in \widetilde{\mathcal{X}}_i} \mathbb{1}_{h(\boldsymbol{x}) \neq H_{t-1}(\boldsymbol{x})}$, $\widehat{\epsilon}_{\mathcal{D}_t}(h, H_{t-1}) = \frac{1}{N_t} \sum_{\boldsymbol{x} \in \mathcal{X}_i} \mathbb{1}_{h(\boldsymbol{x}) \neq H_{t-1}(\boldsymbol{x})}$, and $\Omega \triangleq \{\alpha_i, \beta_i, \gamma_i\}_{i=1}^{t-1}$.*

Theorem 3.4 offers the opportunity of adaptively adjusting the coefficients ($\alpha_i$, $\beta_i$, and $\gamma_i$) according to the data (current domain data $\mathcal{S}_t$ and the memory bank $\mathcal{M} = \{M_i\}_{i=1}^{t-1}$) and history model ($H_{t-1}$) at hand, thereby achieving the tightest bound. For example, when the $\mathcal{H}\Delta\mathcal{H}$ divergence between domain $i$ and domain $t$, i.e., $d_{\mathcal{H}\Delta\mathcal{H}}(\mathcal{D}_i, \mathcal{D}_t)$, is small, minimizing this unified bound (Eqn. 8) leads to a large coefficient $\beta_i$ and therefore naturally puts on more focus on cross-domain bound in Eqn. 35 which leverages the current domain $t$'s data to estimate the true risk.

**UDIL as a Unified Framework.** Interestingly, Eqn. 8 unifies various domain incremental learning methods. Table 1 shows that different methods are equivalent to fixing the coefficients $\{\alpha_i, \beta_i, \gamma_i\}_{i=1}^{t-1}$ to different values (see Appendix B for a detailed discussion). For example, assuming default configurations, LwF [52] corresponds to Eqn. 8 with *fixed* coefficients $\{\alpha_i = \gamma_i = 0, \beta_i = 1\}$; ER [75] corresponds to Eqn. 8 with *fixed* coefficients $\{\alpha_i = \beta_i = 0, \gamma_i = 1\}$, and DER++ [8] corresponds to Eqn. 8 with *fixed* coefficients $\{\alpha_i = \gamma_i = 0.5, \beta_i = 0\}$, under certain achievable conditions. Inspired by this unification, our UDIL adaptively adjusts these coefficients to search for the tightest bound in the range $[0, 1]$ when each domain arrives during domain incremental learning, thereby improving performance. Corollary 3.4.1 below shows that such *adaptive* bound is always tighter, or at least as tight as, any bounds with *fixed* coefficients.

**Corollary 3.4.1.** *For any bound $g(h, H_{t-1}, \Omega_{fixed})$ (defined in Eqn. 8) with* fixed *coefficients $\Omega_{fixed}$, e.g., $\Omega_{fixed} = \Omega_{ER} = \{\alpha_i = \beta_i = 0, \gamma_i = 1\}_{i=1}^{t-1}$ for ER [75], we have*

$$
\sum_{i=1}^{t} \epsilon_{\mathcal{D}_i}(h) \leq \min_{\Omega} g(h, H_{t-1}, \Omega) \leq g(h, H_{t-1}, \Omega_{fixed}), \quad \forall h, H_{t-1} \in \mathcal{H}. \tag{9}
$$

Corollary 3.4.1 shows that the unified bound Eqn. 8 with *adaptive* coefficients is always preferable to other bounds with *fixed* coefficients. We therefore use it to develop a better domain incremental learning algorithm in Sec. 4 below.

# 4 Method: Adaptively Minimizing the Tightest Bound in UDIL

Although Theorem 3.4 provides a unified perspective for domain incremental learning, it does not immediately translate to a practical objective function to train a model. It is also unclear what coefficients $\Omega$ for Eqn. 8 would be the best choice. In fact, a *static* and *fixed* setting will not suffice, as different problems may involve different sequences of domains with dynamic changes; therefore ideally $\Omega$ should be *dynamic* (e.g., $\alpha_i \neq \alpha_{i+1}$) and *adaptive* (i.e., learnable from data). In this section, we start by mapping the unified bound in Eqn. 8 to concrete loss terms, discuss how the coefficients $\Omega$ are learned, and then provide a final objective function to learn the optimal model.

## 4.1 From Theory to Practice: Translating the Bound in Eqn. 8 to Differentiable Loss Terms

**(1) ERM Terms.** We use the cross-entropy classification loss in Definition 4.1 below to optimize domain $t$'s ERM term $\widehat{\epsilon}_{\mathcal{D}_t}(h)$ and memory replay ERM terms $\{\gamma_i \widehat{\epsilon}_{\mathcal{D}_i}(h)\}_{i=1}^{t-1}$ in Eqn. 8.

**Definition 4.1** (**Classification Loss**). *Let $h : \mathbb{R}^n \to \mathbb{S}^{K-1}$ be a function that maps the input $\boldsymbol{x} \in \mathbb{R}^n$ to the space of $K$-class probability simplex, i.e., $\mathbb{S}^{K-1} \triangleq \{\boldsymbol{z} \in \mathbb{R}^K : z_i \geq 0, \sum_i z_i = 1\}$; let $\mathcal{X}$ be a collection of samples drawn from an arbitrary data distribution and $f : \mathbb{R}^n \to [K]$ be the function that maps the input to the true label. The classification loss is defined as the average cross-entropy between the true label $f(\boldsymbol{x})$ and the predicted probability $h(\boldsymbol{x})$, i.e.,*

$$\widehat{\ell}_{\mathcal{X}}(h) \triangleq \frac{1}{|\mathcal{X}|} \sum_{\boldsymbol{x} \in \mathcal{X}} \left[ -\sum_{j=1}^{K} \mathbb{1}_{f(\boldsymbol{x})=j} \cdot \log\left([h(\boldsymbol{x})]_j\right) \right]. \tag{10}$$

Following Definition 4.1, we replace $\widehat{\epsilon}_{\mathcal{D}_t}(h)$ and $\widehat{\epsilon}_{\mathcal{D}_i}(h)$ in Eqn. 8 with $\widehat{\ell}_{\mathcal{X}_t}(h)$ and $\widehat{\ell}_{\mathcal{X}_i}(h)$.

**(2) Intra- and Cross-Domain Terms.** We use the distillation loss below to optimize intra-domain ($\{\widehat{\epsilon}_{\mathcal{D}_i}(h, H_{t-1})\}_{i=1}^{t-1}$) and cross-domain ($\widehat{\epsilon}_{\mathcal{D}_t}(h, H_{t-1})$) model-based error terms in Eqn. 8.

**Definition 4.2** (**Distillation Loss**). *Let $h, H_{t-1} : \mathbb{R}^n \to \mathbb{S}^{K-1}$ both be functions that map the input $\boldsymbol{x} \in \mathbb{R}^n$ to the space of $K$-class probability simplex as defined in Definition 4.1; let $\mathcal{X}$ be a collection of samples drawn from an arbitrary data distribution. The distillation loss is defined as the average cross-entropy between the target probability $H_{t-1}(\boldsymbol{x})$ and the predicted probability $h(\boldsymbol{x})$, i.e.,*

$$\widehat{\ell}_{\mathcal{X}}(h, H_{t-1}) \triangleq \frac{1}{|\mathcal{X}|} \sum_{\boldsymbol{x} \in \mathcal{X}} \left[ -\sum_{j=1}^{K} [H_{t-1}(\boldsymbol{x})]_j \cdot \log\left([h(\boldsymbol{x})]_j\right) \right]. \tag{11}$$

Accordingly, we replace $\widehat{\epsilon}_{\mathcal{D}_i}(h, H_{t-1})$ with $\widehat{\ell}_{\mathcal{X}_i}(h, H_{t-1})$ and $\widehat{\epsilon}_{\mathcal{D}_t}(h, H_{t-1})$ with $\widehat{\ell}_{\mathcal{X}_t}(h, H_{t-1})$.

**(3) Constant Term.** The error term $\sum_{i=1}^{t-1}(\alpha_i + \beta_i)\epsilon_{\mathcal{D}_i}(H_{t-1})$ in Eqn. 8 is a constant and contains no trainable parameters (since $H_{t-1}$ is a fixed history model); therefore it does not need a loss term.

**(4) Divergence Term.** In Eqn. 8, $\sum_{i=1}^{t-1} \beta_i d_{\mathcal{H}\Delta\mathcal{H}}(\mathcal{D}_i, \mathcal{D}_t)$ measures the weighted average of the dissimilarity between domain $i$'s and domain $t$'s data distributions. Inspired by existing adversarial domain adaptation methods [57, 26, 109, 17, 102, 101, 94], we can further tighten this divergence term by considering the *embedding distributions* instead of *data distributions* using an learnable encoder. Specifically, given an encoder $e : \mathbb{R}^n \to \mathbb{R}^m$ and a family of domain discriminators (classifiers) $\mathcal{H}_d$, we have the empirical estimate of the divergence term as follows:

$$\sum_{i=1}^{t-1} \beta_i \widehat{d}_{\mathcal{H}\Delta\mathcal{H}}(e(\mathcal{U}_i), e(\mathcal{U}_t)) = 2 \sum_{i=1}^{t-1} \beta_i - 2 \min_{d \in \mathcal{H}_d} \sum_{i=1}^{t-1} \beta_i \left[ \frac{1}{|\mathcal{U}_i|} \sum_{\boldsymbol{x} \in \mathcal{U}_i} \mathbb{1}_{\Delta_i(\boldsymbol{x}) \geq 0} + \frac{1}{|\mathcal{U}_t|} \sum_{\boldsymbol{x} \in \mathcal{U}_t} \mathbb{1}_{\Delta_i(\boldsymbol{x}) < 0} \right],$$

where $\mathcal{U}_i$ (and $\mathcal{U}_t$) is a set of samples drawn from domain $\mathcal{D}_i$ (and $\mathcal{D}_t$), $d : \mathbb{R}^m \to \mathbb{S}^{t-1}$ is a domain classifier, and $\Delta_i(\boldsymbol{x}) = [d(e(\boldsymbol{x}))]_i - [d(e(\boldsymbol{x}))]_t$ is the difference between the probability of $\boldsymbol{x}$ belonging to domain $i$ and domain $t$. Replacing the indicator function with the differentiable cross-entropy loss, $\sum_{i=1}^{t-1} \beta_i \widehat{d}_{\mathcal{H}\Delta\mathcal{H}}(e(\mathcal{U}_i), e(\mathcal{U}_t))$ above then becomes

$$2 \sum_{i=1}^{t-1} \beta_i - 2 \min_{d \in \mathcal{H}_d} \sum_{i=1}^{t-1} \beta_i \left[ \frac{1}{N_i} \sum_{\boldsymbol{x} \in \mathcal{X}_i} [-\log\left([d(e(\boldsymbol{x}))]_i\right)] + \frac{1}{N_t} \sum_{\boldsymbol{x} \in \mathcal{S}_t} [-\log\left([d(e(\boldsymbol{x}))]_t\right)] \right]. \tag{12}$$

**Algorithm 1** Unified Domain Incremental Learning (UDIL) for Domain $t$ Training

**Require:** history model $H_{t-1} = P_{t-1} \circ E_{t-1}$, current model $h_\theta = p \circ e$, discriminator model $d_\phi$;
**Require:** dataset from the current domain $\mathcal{S}_t$, memory bank $\mathcal{M} = \{M_i\}_{i=1}^{t-1}$;
**Require:** training steps $S$, batch size $B$, learning rate $\eta$;
**Require:** domain alignment strength coefficient $\lambda_d$, hyperparameter for generalization effect $C$.

1: $h_\theta \leftarrow H_{t-1}$          ▷ Initialization of the current model.
2: $\Omega \triangleq \{\alpha_i, \beta_i, \gamma_i\} \leftarrow \{1/3, 1/3, 1/3\}$, for $\forall i \in [t-1]$    ▷ Initialization of the replay coefficient $\Omega$.
3: **for** $s = 1, \cdots, S$ **do**
4:      $B_t \sim \mathcal{S}_t; B_i \sim M_i, \forall i \in [t-1]$      ▷ Sample a mini-batch of data from all domains.
5:      $\phi \leftarrow \phi - \eta \cdot \lambda_d \cdot \nabla_\phi V_d(d, e, \mathring{\Omega})$      ▷ Discriminator training with Eqn. 16.
6:      $\Omega \leftarrow \Omega - \eta \cdot \nabla_\Omega V_{0\text{-}1}(\mathring{h}, \Omega)$      ▷ Find a tighter bound with Eqn. 15.
7:      $\theta \leftarrow \theta - \eta \cdot \nabla_\theta (V_l(h_\theta, \mathring{\Omega}) - \lambda_d V_d(d, e, \mathring{\Omega}))$    ▷ Model training with Eqn. 14 and Eqn. 16.
8: **end for**
9: $H_t \leftarrow h$
10: $\mathcal{M} \leftarrow \text{BalancedSampling}(\mathcal{M}, \mathcal{S}_t)$
11: **return** $H_t$      ▷ For training on domain $t+1$.

## 4.2 Putting Everything Together: UDIL Training Algorithm

**Objective Function.** With these differentiable loss terms above, we can derive an algorithm that learns the optimal model by minimizing the tightest bound in Eqn. 8. As mentioned above, to achieve a tighter $d_{\mathcal{H}\Delta\mathcal{H}}$, we decompose the hypothesis as $h = p \circ e$, where $e : \mathbb{R}^n \to \mathbb{R}^m$ and $p : \mathbb{R}^m \to \mathbb{S}^{K-1}$ are the encoder and predictor, respectively. To find and to minimize the tightest bound in Theorem 3.4, we treat $\Omega = \{\alpha_i, \beta_i, \gamma_i\}_{i=1}^{t-1}$ as learnable parameters and seek to optimize the following objective (we denote as $\mathring{x} = \text{sg}(x)$ the 'copy-weights-and-stop-gradients' operation):

$$\min_{\{\Omega, h=p\circ e\}} \max_{d} \quad V_l(h, \mathring{\Omega}) + V_{0\text{-}1}(\mathring{h}, \Omega) - \lambda_d V_d(d, e, \mathring{\Omega}) \tag{13}$$
$$\text{s.t.} \quad \alpha_i + \beta_i + \gamma_i = 1, \quad \forall i \in \{1, 2, \ldots, t-1\}$$
$$\alpha_i, \beta_i, \gamma_i \geq 0, \quad \forall i \in \{1, 2, \ldots, t-1\}$$

**Details of $V_l$, $V_{0\text{-}1}$, and $V_d$.** $V_l$ is the loss for **learning the model** $h$, where the terms $\widehat{\ell}.(\cdot)$ are *differentiable cross-entropy losses* as defined in Eqn. 10 and Eqn. 11:

$$V_l(h, \mathring{\Omega}) = \sum\nolimits_{i=1}^{t-1} \left[ \mathring{\gamma}_i \widehat{\ell}_{\mathcal{X}_i}(h) + \mathring{\alpha}_i \widehat{\ell}_{\mathcal{X}_i}(h, H_{t-1}) \right] + \widehat{\ell}_{\mathcal{S}_t}(h) + \left(\sum\nolimits_{i=1}^{t-1} \mathring{\beta}_i\right)\widehat{\ell}_{\mathcal{S}_t}(h, H_{t-1}). \tag{14}$$

$V_{0\text{-}1}$ is the loss for **finding the optimal coefficient set** $\Omega$. Its loss terms use Definition 3.1 and Eqn. 12 to estimate ERM terms and $\mathcal{H}\Delta\mathcal{H}$-divergence, respectively:

$$V_{0\text{-}1}(\mathring{h}, \Omega) = \sum\nolimits_{i=1}^{t-1} \left[ \gamma_i \widehat{\epsilon}_{\mathcal{D}_i}(\mathring{h}) + \alpha_i \widehat{\epsilon}_{\mathcal{D}_i}(\mathring{h}, H_{t-1}) \right] + \left(\sum\nolimits_{i=1}^{t-1} \beta_i\right)\widehat{\epsilon}_{\mathcal{D}_t}(\mathring{h}, H_{t-1})$$
$$+ \frac{1}{2} \sum\nolimits_{i=1}^{t-1} \beta_i \widehat{d}_{\mathcal{H}\Delta\mathcal{H}}\left(\mathring{e}(\mathcal{X}_i), \mathring{e}(\mathcal{S}_t)\right) + \sum\nolimits_{i=1}^{t-1} (\alpha_i + \beta_i)\widehat{\epsilon}_{\mathcal{D}_i}(H_{t-1})$$
$$+ C \cdot \sqrt{\left(\frac{(1 + \sum_{i=1}^{t-1} \beta_i)^2}{N_t} + \sum\nolimits_{i=1}^{t-1} \frac{(\gamma_i + \alpha_i)^2}{\widetilde{N}_i}\right)}. \tag{15}$$

In Eqn. 15, $\widehat{\epsilon}.(\cdot)$ uses *discrete 0-1 loss*, which is different from Eqn. 14, and a hyper-parameter $C = \sqrt{8d\log(2eN/d) + 8\log(2/\delta)}$ is introduced to model the combined influence of $\mathcal{H}$'s VC-dimension and $\delta$.

$V_d$ follows Eqn. 12 to **minimize the divergence between different domains' embedding distributions** (i.e., aligning domains) by the minimax game between $e$ and $d$ with the value function:

$$V_d(d, e, \mathring{\Omega}) = \left(\sum\nolimits_{i=1}^{t-1} \mathring{\beta}_i\right) \frac{1}{N_i} \sum_{\boldsymbol{x} \in \mathcal{S}_t} [-\log([d(e(\boldsymbol{x}))]_t)] + \sum_{i=1}^{t-1} \frac{\mathring{\beta}_i}{\widetilde{N}_i} \sum_{\boldsymbol{x} \in \widetilde{\mathcal{X}}_i} [-\log([d(e(\boldsymbol{x}))]_i)]. \tag{16}$$

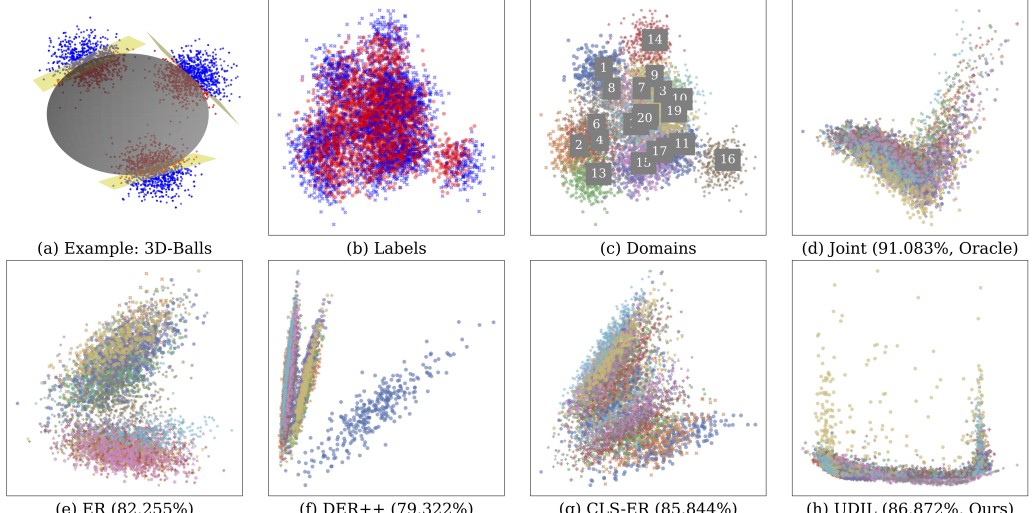

| (a) Example: 3D-Balls | (b) Labels | (c) Domains | (d) Joint (91.083%, Oracle) |
| (e) ER (82.255%) | (f) DER++ (79.322%) | (g) CLS-ER (85.844%) | (h) UDIL (86.872%, Ours) |

Figure 1: Results on *HD-Balls*. In (a-b), data is colored according to labels; in (c-h), data is colored according to domain ID. All data is plotted after PCA [6]. **(a)** Simplified *HD-Balls* dataset with 3 domains in the 3D space (for visualization purposes only). **(b-c)** Embeddings of *HD-Balls*'s raw data colored by labels and domain ID. **(d-h)** Accuracy and embeddings learned by Joint (oracle), UDIL, and three best baselines (more in Appendix C.5). Joint, as the *oracle*, naturally aligns different domains, and UDIL outperforms all baselines in terms of embedding alignment and accuracy.

Here in Eqn. 16, if an optimal $d^*$ and a fixed $\Omega$ is given, maximizing $V_d(d^*, e, \Omega)$ with respect to the encoder $e$ is equivalent to minimizing the weighted sum of the divergence $\sum_{i=1}^{t-1} \beta_i d_{\mathcal{H}\Delta\mathcal{H}}(e(\mathcal{D}_i), e(\mathcal{D}_t))$. This result indicates that the divergence between two domains' *embedding distributions* can be actually minimized. Intuitively this minimax game learns an encoder $e$ that aligns the embedding distributions of different domains so that their domain IDs can not be predicted (distinguished) by a powerful discriminator given an embedding $e(\boldsymbol{x})$. Algorithm 1 below outlines how UDIL minimizes the tightest bound. Please refer to Appendix C for more implementation details, including a model diagram in Fig. 2.

## 5 Experiments

In this section, we compare UDIL with existing methods on both synthetic and real-world datasets.

### 5.1 Baselines and Implementation Details

We compare UDIL with the state-of-the-art continual learning methods that are either specifically designed for domain incremental learning or can be easily adapted to the domain incremental learning setting. For fair comparison, we do not consider methods that leverage large-scale pre-training or prompt-tuning [99, 98, 53, 88]. Exemplar-free baselines include online Elastic Weight Consolidation (**oEWC**) [81], Synaptic Intelligence (**SI**) [105], and Learning without Forgetting (**LwF**) [52]. Memory-based domain incremental learning baselines include Gradient Episodic Memory (**GEM**) [58], Averaged Gradient Episodic Memory (**A-GEM**) [10], Experience Replay (**ER**) [75], Dark Experience Replay (**DER++**) [8], and two recent methods, Complementary Learning System based Experience Replay (**CLS-ER**) [4] and Error Sensitivity Modulation based Experience Replay (**ESM-ER**) [80] (see Appendix C.5 for more detailed introduction to the baseline methods above). In addition, we implement the fine-tuning (**Fine-tune**) [52] and joint-training (**Joint**) as the performance lower bound and upper bound (Oracle).

We train all models using three different random seeds and report the mean and standard deviation. All methods are implemented with PyTorch [70], based on the mammoth code base [7, 8], and run on a single NVIDIA RTX A5000 GPU. For fair comparison, within the same dataset, all methods adopt the same neural network architecture, and the memory sampling strategy is set to random

Table 2: **Performances (%) on *HD-Balls*, *P-MNIST*, and *R-MNIST*.** We use two metrics, Average Accuracy and Forgetting, to evaluate the methods' effectiveness. "↑" and "↓" mean higher and lower numbers are better, respectively. We use **boldface** and underlining to denote the best and the second-best performance, respectively. We use "-" to denote "not appliable".

| Method | Buffer | *HD-Balls* | | *P-MNIST* | | *R-MNIST* | |
|---|---|---|---|---|---|---|---|
| | | Avg. Acc (↑) | Forgetting (↓) | Avg. Acc (↑) | Forgetting (↓) | Avg. Acc (↑) | Forgetting (↓) |
| Fine-tune | - | $52.319_{\pm0.024}$ | $43.520_{\pm0.079}$ | $70.102_{\pm2.945}$ | $27.522_{\pm3.042}$ | $47.803_{\pm1.703}$ | $52.281_{\pm1.797}$ |
| oEWC [81] | - | $54.131_{\pm0.193}$ | $39.743_{\pm1.388}$ | $78.476_{\pm1.223}$ | $18.068_{\pm1.321}$ | $48.203_{\pm0.827}$ | $51.181_{\pm0.867}$ |
| SI [105] | - | $52.303_{\pm0.037}$ | $43.175_{\pm0.041}$ | $79.045_{\pm1.357}$ | $17.409_{\pm1.446}$ | $48.251_{\pm1.381}$ | $51.053_{\pm1.507}$ |
| LwF [52] | - | $51.523_{\pm0.065}$ | $25.155_{\pm0.264}$ | $73.545_{\pm2.646}$ | $24.556_{\pm2.789}$ | $54.709_{\pm0.515}$ | $45.473_{\pm0.565}$ |
| GEM [58] | | $69.747_{\pm0.656}$ | $13.591_{\pm0.779}$ | $89.097_{\pm0.149}$ | $6.975_{\pm0.167}$ | $76.619_{\pm0.581}$ | $21.289_{\pm0.579}$ |
| A-GEM [10] | | $62.777_{\pm0.295}$ | $12.878_{\pm1.588}$ | $87.560_{\pm0.087}$ | $8.577_{\pm0.053}$ | $59.654_{\pm0.122}$ | $39.196_{\pm0.171}$ |
| ER [75] | | $82.255_{\pm1.552}$ | $9.524_{\pm1.655}$ | $88.339_{\pm0.044}$ | $7.180_{\pm0.029}$ | $76.794_{\pm0.696}$ | $20.696_{\pm0.744}$ |
| DER++ [8] | 400 | $79.332_{\pm1.347}$ | $13.762_{\pm1.514}$ | $\mathbf{92.950_{\pm0.361}}$ | $3.378_{\pm0.245}$ | $84.258_{\pm0.544}$ | $13.692_{\pm0.560}$ |
| CLS-ER [4] | | $85.844_{\pm0.165}$ | $5.297_{\pm0.281}$ | $91.598_{\pm0.117}$ | $3.795_{\pm0.144}$ | $81.771_{\pm0.354}$ | $15.455_{\pm0.356}$ |
| ESM-ER [80] | | $71.995_{\pm3.833}$ | $13.245_{\pm5.397}$ | $89.829_{\pm0.698}$ | $6.888_{\pm0.738}$ | $82.192_{\pm0.164}$ | $16.195_{\pm0.150}$ |
| UDIL (Ours) | | $\mathbf{86.872_{\pm0.195}}$ | $\mathbf{3.428_{\pm0.359}}$ | $92.666_{\pm0.108}$ | $\mathbf{2.853_{\pm0.107}}$ | $\mathbf{86.635_{\pm0.686}}$ | $\mathbf{8.506_{\pm1.181}}$ |
| Joint (Oracle) | ∞ | $91.083_{\pm0.332}$ | - | $96.368_{\pm0.042}$ | - | $97.150_{\pm0.036}$ | - |

balanced sampling (see Appendix C.2 and Appendix C.6 for more implementation details on training). We evaluate all methods with standard continual learning metrics including 'average accuracy', 'forgetting', and 'forward transfer' (see Appendix C.4 for detailed definitions).

## 5.2 Toy Dataset: High-Dimensional Balls

To gain insight into UDIL, we start with a toy dataset, high dimensional balls on a sphere (referred to as *HD-Balls* below), for domain incremental learning. *HD-Balls* includes 20 domains, each containing 2,000 data points sampled from a Gaussian distribution $\mathcal{N}(\boldsymbol{\mu}, 0.2^2\boldsymbol{I})$. The mean $\boldsymbol{\mu}$ is randomly sampled from a 100-dimensional unit sphere, i.e., $\{\boldsymbol{\mu} \in \mathbb{R}^{100} : \|\boldsymbol{\mu}\|_2 = 1\}$; the covariance matrix $\Sigma$ is fixed. In *HD-Balls*, each domain represents a binary classification task, where the decision boundary is the hyperplane that passes the center $\boldsymbol{\mu}$ and is tangent to the unit sphere. Fig. 1(a-c) shows some visualization on *HD-Balls*.

Column 3 and 4 of Table 2 compare the performance of our UDIL with different baselines. We can see that UDIL achieves the highest final average accuracy and the lowest forgetting. Fig. 1(d-h) shows the embedding distributions (i.e., $e(\boldsymbol{x})$) for different methods. We can see better embedding alignment across domains generally leads to better performance. Specifically, Joint, as the oracle, naturally aligns different domains' embedding distributions and achieves an accuracy upper bound of $91.083\%$. Similarly, our UDIL can adaptively adjust the coefficients of different loss terms, including Eqn. 12, successfully align different domains, and thereby outperform all baselines.

## 5.3 Permutation MNIST

We further evaluate our method on the Permutation MNIST (*P-MNIST*) dataset [50]. *P-MNIST* includes 20 sequential domains, with each domain constructed by applying a fixed random permutation to the pixels in the images. Column 5 and 6 of Table 2 show the results of different methods. Our UDIL achieves the second best ($92.666\%$) final average accuracy, which is only $0.284\%$ lower than the best baseline DER++. We believe this is because (i) there is not much space for improvement as the gap between joint-training (oracle) and most methods are small; (ii) under the permutation, different domains' data distributions are too distinct from each other, lacking the meaningful relations among the domains, and therefore weakens the effect of embedding alignment in our method. Nevertheless, UDIL still achieves best performance in terms of forgetting ($2.853\%$). This is mainly because our unified UDIL framework (i) is directly derived from the total loss of *all* domains, and (ii) uses adaptive coefficients to achieve a more balanced trade-off between learning the current domain and avoiding forgetting previous domains.

Table 3: **Performances (%) evaluated on *Seq-CORe50*.** We use three metrics, Average Accuracy, Forgetting, and Forward Transfer, to evaluate the methods' effectiveness. "↑" and "↓" mean higher and lower numbers are better, respectively. We use **boldface** and underlining to denote the best and the second-best performance, respectively. We use "-" to denote "not appliable" and "⋆" to denote out-of-memory (*OOM*) error when running the experiments.

| Method | Buffer | $\mathcal{D}_{1:3}$ | $\mathcal{D}_{4:6}$ | $\mathcal{D}_{7:9}$ | $\mathcal{D}_{10:11}$ | Avg. Acc (↑) | Overall Forgetting (↓) | Fwd. Transfer (↑) |
|---|---|---|---|---|---|---|---|---|
| | | Avg. Acc (↑) | | | | | | |
| Fine-tune | - | $73.707_{\pm13.144}$ | $34.551_{\pm1.254}$ | $29.406_{\pm2.579}$ | $28.689_{\pm3.144}$ | $31.832_{\pm1.034}$ | $73.296_{\pm1.399}$ | $15.153_{\pm0.255}$ |
| oEWC [81] | - | $74.567_{\pm13.360}$ | $35.915_{\pm0.260}$ | $30.174_{\pm3.195}$ | $28.291_{\pm2.522}$ | $30.813_{\pm1.154}$ | $74.563_{\pm0.937}$ | $15.041_{\pm0.249}$ |
| SI [105] | - | $74.661_{\pm14.162}$ | $34.345_{\pm1.001}$ | $30.127_{\pm2.971}$ | $28.839_{\pm3.631}$ | $32.469_{\pm1.315}$ | $73.144_{\pm1.588}$ | $14.837_{\pm1.005}$ |
| LwF [52] | - | $80.383_{\pm10.190}$ | $28.357_{\pm1.143}$ | $31.386_{\pm0.787}$ | $28.711_{\pm2.981}$ | $31.692_{\pm0.768}$ | $72.990_{\pm1.350}$ | $15.356_{\pm0.750}$ |
| GEM [58] | | $79.852_{\pm6.864}$ | $38.961_{\pm1.718}$ | $39.258_{\pm2.614}$ | $36.859_{\pm0.842}$ | $37.701_{\pm0.273}$ | $22.724_{\pm1.554}$ | $19.030_{\pm0.936}$ |
| A-GEM [10] | | $80.348_{\pm9.394}$ | $41.472_{\pm3.394}$ | $43.213_{\pm1.542}$ | $39.181_{\pm3.999}$ | $43.181_{\pm2.025}$ | $33.775_{\pm3.003}$ | $19.033_{\pm0.792}$ |
| ER [75] | | $90.838_{\pm2.177}$ | $79.343_{\pm2.699}$ | $68.151_{\pm0.226}$ | $65.034_{\pm1.571}$ | $66.605_{\pm0.214}$ | $32.750_{\pm0.455}$ | $21.735_{\pm0.802}$ |
| DER++ [8] | 500 | $92.444_{\pm1.764}$ | $88.652_{\pm1.854}$ | $80.391_{\pm0.107}$ | $\underline{78.038}_{\pm0.591}$ | $\underline{78.629}_{\pm0.753}$ | $\underline{21.910}_{\pm1.094}$ | $\underline{22.488}_{\pm1.049}$ |
| CLS-ER [4] | | $89.834_{\pm1.323}$ | $78.909_{\pm1.724}$ | $70.591_{\pm0.322}$ | $\star$ | $\star$ | $\star$ | $\star$ |
| ESM-ER [80] | | $84.905_{\pm6.471}$ | $51.905_{\pm3.257}$ | $53.815_{\pm1.770}$ | $50.178_{\pm2.574}$ | $52.751_{\pm1.296}$ | $25.444_{\pm0.580}$ | $21.435_{\pm1.018}$ |
| UDIL (Ours) | | $\mathbf{98.152}_{\pm1.665}$ | $\mathbf{89.814}_{\pm2.302}$ | $\mathbf{83.052}_{\pm0.151}$ | $\mathbf{81.547}_{\pm0.269}$ | $\mathbf{82.103}_{\pm0.279}$ | $\mathbf{19.589}_{\pm0.303}$ | $\mathbf{31.215}_{\pm0.831}$ |
| Joint (Oracle) | $\infty$ | - | - | - | - | $99.137_{\pm0.049}$ | - | - |

## 5.4 Rotating MNIST

We also evaluate our method on the Rotating MNIST dataset (*R-MNIST*) containing 20 sequential domains. Different from *P-MNIST* where shift from domain $t$ to domain $t + 1$ is abrupt, *R-MNIST*'s domain shift is gradual. Specifically, domain $t$'s images are rotated by an angle randomly sampled from the range $[9° \cdot (t − 1), 9° \cdot t)$. Column 7 and 8 of Table 2 show that our UDIL achieves the highest average accuracy (86.635%) and the lowest forgetting (8.506%) simultaneously, significantly improving on the best baseline DER++ (average accuracy of 84.258% and forgetting of 13.692%). Interestingly, such improvement is achieved when our UDIL's $\beta_i$ is high, which further verifies that UDIL indeed leverages the similarities shared across different domains so that the generalization error is reduced.

## 5.5 Sequential CORe50

CORe50 [55, 56] is a real-world continual object recognition dataset that contains 50 domestic objects collected from 11 domains (120,000 images in total). Prior work has used CORe50 for settings such as domain generalization (e.g., train a model on only 8 domains and test it on 3 domains), which is different from our domain-incremental learning setting. To focus the evaluation on alleviating catastrophic forgetting, we retain 20% of the data as the test set and continually train the model on these 11 domains; we therefore call this dataset variant *Seq-CORe50*. Table 3 shows that our UDIL outperforms all baselines in every aspect on *Seq-CORe50*. Besides the average accuracy over all domains, we also report average accuracy over different domain intervals (e.g., $\mathcal{D}_{1:3}$ denotes average accuracy from domain 1 to domain 3) to show how different model's performance drops over time. The results show that our UDIL consistently achieves the highest average accuracy until the end. It is also worth noting that UDIL also achieves the best performance on another two metrics, i.e., forgetting and forward transfer.

## 6 Conclusion

In this paper, we propose a principled framework, UDIL, for domain incremental learning with memory to unify various existing methods. Our theoretical analysis shows that different existing methods are equivalent to minimizing the same error bound with different *fixed* coefficients. With this unification, our UDIL allows *adaptive* coefficients during training, thereby always achieving the tightest bound and improving the performance. Empirical results show that our UDIL outperforms the state-of-the-art domain incremental learning methods on both synthetic and real-world datasets. One limitation of this work is the implicit *i.i.d.* exemplar assumption, which may not hold if memory is selected using specific strategies. Addressing this limitation can lead to a more powerful unified framework and algorithms, which would be interesting future work.

# Acknowledgement

The authors thank the reviewers/AC for the constructive comments to improve the paper. HS and HW are partially supported by NSF Grant IIS-2127918. The views and conclusions contained herein are those of the authors and should not be interpreted as necessarily representing the official policies, either expressed or implied, of the sponsors.

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

# Appendix

In Sec. A, we present the proofs for the lemmas, theorems, and corollaries presented in the main body of our work. Sec. B discusses the correspondence of existing methods to specific cases within our framework. In Sec. C, we provide a detailed presentation of our final algorithm, UDIL, including an algorithmic description, a visual diagram, and implementation details. We introduce the experimental settings, including the evaluation metrics and specific training schemes. Finally, in Sec. D, we present additional empirical results with varying memory sizes and provide more visualization results.

## A Proofs of Lemmas, Theorems, and Corollaries

Before proceeding to prove any lemmas or theorems, we first introduce three crucial additional lemmas that will be utilized in the subsequent sections. Among these, Lemma A.1 offers a generalization bound for any weighted summation of ERM losses across multiple domains. Furthermore, Lemma A.2 provides a generalization bound for a weighted summation of *labeling functions* within a given domain. Lastly, we highlight Lemma 3 in [5] as Lemma A.3, which will be used to establish the upper bound for Lemma 3.3.

**Lemma A.1** (**Generalization Bound of $\alpha$-weighted Domains**). *Let $\mathcal{H}$ be a hypothesis space of VC dimension $d$. Assume $N_j$ denotes the number of the samples collected from domain $j$, and $N = \sum_j N_j$ is the total number of the examples collected from all domains. Then for any $\alpha_j > 0$ and $\delta \in (0, 1)$, with probability at least $1 - \delta$:*

$$\sum_j \alpha_j \epsilon_{\mathcal{D}_j}(h) \leq \sum_j \alpha_j \widehat{\epsilon}_{\mathcal{D}_j}(h) + \sqrt{\left(\sum_j \frac{\alpha_j^2}{N_j}\right)\left(8d \log\left(\frac{2eN}{d}\right) + 8\log\left(\frac{2}{\delta}\right)\right)}. \tag{17}$$

*Proof.* Suppose each domain $\mathcal{D}_j$ has a deterministic ground-truth labeling function $f_j : \mathbb{R}^n \to \{0, 1\}$. Denote as $\widehat{\epsilon}_\alpha \triangleq \sum_j \alpha_j \widehat{\epsilon}_{\mathcal{D}_j}(h)$ the $\alpha$-weighted empirical loss evaluated on different domains. Hence,

$$\widehat{\epsilon}_\alpha(h) = \sum_j \alpha_j \widehat{\epsilon}_{\mathcal{D}_j}(h) = \sum_j \alpha_j \frac{1}{N_j} \sum_{\boldsymbol{x} \in \mathcal{X}_j} \mathbb{1}_{h(\boldsymbol{x}) \neq f_j(\boldsymbol{x})} = \frac{1}{N} \sum_j \sum_{k=1}^{N_j} R_{j,k}, \tag{18}$$

where $R_{j,k} = \left(\frac{\alpha_j N_j}{N}\right) \cdot \mathbb{1}_{h(\boldsymbol{x}_k) \neq f_j(\boldsymbol{x}_k)}$ is a random variable that takes the values in $\{\frac{\alpha_j N_j}{N}, 0\}$. By the linearity of the expectation, we have $\epsilon_\alpha(h) = \mathbb{E}[\widehat{\epsilon}_\alpha(h)]$. Following [3, 62], we have

$$\mathbb{P}\left\{\exists h \in \mathcal{H}, \text{s.t. } |\widehat{\epsilon}_\alpha(h) - \epsilon_\alpha(h)| \geq \epsilon\right\} \tag{19}$$

$$\leq \quad 2 \cdot \mathbb{P}\left\{\sup_{h \in \mathcal{H}} |\widehat{\epsilon}_\alpha(h) - \widehat{\epsilon}_\alpha'(h)| \geq \frac{\epsilon}{2}\right\} \tag{20}$$

$$\leq \quad 2 \cdot \mathbb{P}\left\{\bigcup_{R_{j,k}, R_{j,k}'} \frac{1}{N}\left|\sum_j \sum_{k=1}^{N_j}(R_{j,k} - R_{j,k}')\right| \geq \frac{\epsilon}{2}\right\} \tag{21}$$

$$\leq \quad 2\Pi_{\mathcal{H}}(2N) \exp\left\{\frac{-2(N\epsilon/2)^2}{\sum_j(N_j)(2\alpha_j N/N_j)^2}\right\} \tag{22}$$

$$= \quad 2\Pi_{\mathcal{H}}(2N) \exp\left\{-\frac{\epsilon^2}{8\sum_j(\alpha_j^2/N_j)}\right\} \tag{23}$$

$$\leq \quad 2(2N)^d \exp\left\{-\frac{\epsilon^2}{8\sum_j(\alpha_j^2/N_j)}\right\}, \tag{24}$$

where in Eqn. 20, $\widehat{\epsilon}_\alpha'(h)$ is the $\alpha$-weighted empirical loss evaluated on the "ghost" set of examples $\{\mathcal{X}_j'\}$; Eqn. 22 is yielded by applying Hoeffding's inequalities [37] and introducing the growth function $\Pi_{\mathcal{H}}$ [3, 62, 91] at the same time; Eqn. 24 is achieved by using the fact $\Pi_{\mathcal{H}}(2N) \leq (e \cdot 2N/d)^d \leq (2N)^d$, where $d$ is the VC-dimension of the hypothesis set $\mathcal{H}$. Finally, by setting Eqn. 24 to $\delta$ and solve for the error tolerance $\epsilon$ will complete the proof. $\square$

**Lemma A.2** (**Generalization Bound of $\beta$-weighted Labeling Functions**). *Let $\mathcal{D}$ be a single domain and $\mathcal{X} = \{\boldsymbol{x}_i\}_i^N$ be a collection of samples drawn from $\mathcal{D}$; $\mathcal{H}$ is a hypothesis space of VC dimension*

*d. Suppose $\{f_j : \mathbb{R}^n \to \{0,1\}\}_j$ is a set of different labeling functions. Then for any $\beta_j > 0$ and $\delta \in (0,1)$, with probability at least $1 - \delta$:*

$$\sum_j \beta_j \epsilon_{\mathcal{D}}(h, f_j) \leq \sum_j \beta_j \widehat{\epsilon}_{\mathcal{D}}(h, f_j) + \left(\sum_j \beta_j\right) \sqrt{\tfrac{1}{N} \left(8d \log\left(\tfrac{2eN}{d}\right) + 8 \log\left(\tfrac{2}{\delta}\right)\right)}. \quad (25)$$

*Proof.* Denote as $\epsilon_{\beta}(h) \triangleq \sum_j \beta_j \epsilon_{\mathcal{D}}(h, f_j)$ the $\beta$-weighted error on domain $\mathcal{D}$ and $\{f_j\}_j$ the set of the labeling functions , and $\widehat{\epsilon}_{\beta} \triangleq \sum_j \beta_j \widehat{\epsilon}_{\mathcal{D}}(h, f_j)$ as the $\beta$-weighted empirical loss evaluated on different labeling functions. We have

$$\widehat{\epsilon}_{\beta}(h) = \sum_j \beta_j \frac{1}{N} \sum_{i=1}^{N} \mathbb{1}_{h(\boldsymbol{x}_i) \neq f_j(\boldsymbol{x}_i)} = \frac{1}{N} \sum_{i=1}^{N} \sum_j \beta_j \mathbb{1}_{h(\boldsymbol{x}_i) \neq f_j(\boldsymbol{x}_i)} \triangleq \frac{1}{N} \sum_{i=1}^{N} R_i, \quad (26)$$

where $R_i = \sum_j \beta_j \mathbb{1}_{h(\boldsymbol{x}_i) \neq f_j(\boldsymbol{x}_i)} \in [0, \sum_j \beta_j]$ is a new random variable.

Then we have

$$\mathbb{P}\left\{\exists h \in \mathcal{H}, \text{s.t. } |\widehat{\epsilon}_{\beta}(h) - \epsilon_{\beta}(h)| \geq \epsilon\right\} \quad (27)$$

$$\leq \quad 2 \cdot \mathbb{P}\left\{\sup_{h \in \mathcal{H}} |\widehat{\epsilon}_{\beta}(h) - \widehat{\epsilon}'_{\beta}(h)| \geq \tfrac{\epsilon}{2}\right\} \quad (28)$$

$$\leq \quad 2 \cdot \mathbb{P}\left\{\bigcup_{R_i, R'_i} \frac{1}{N} \left|\sum_{i=1}^{N} (R_i - R'_i)\right| \geq \tfrac{\epsilon}{2}\right\} \quad (29)$$

$$\leq \quad 2\Pi_{\mathcal{H}}(2N) \exp\left\{\frac{-2(N\epsilon/2)^2}{N \cdot (2\sum_j \beta_j)^2}\right\} \quad (30)$$

$$\leq \quad 2(2N)^d \exp\left\{-\frac{N\epsilon^2}{8(\sum_j \beta_j)^2}\right\}, \quad (31)$$

where in Eqn. 28, $\widehat{\epsilon}'_{\beta}(h)$ is the $\beta$-weighted empirical loss evaluated on the "ghost" set of examples $\mathcal{X}'$; Eqn. 30 is yielded by applying Hoeffding's inequalities [37] and introducing the growth function $\Pi_{\mathcal{H}}$ [3, 62, 91] at the same time; Eqn. 31 is achieved by using the fact $\Pi_{\mathcal{H}}(2N) \leq \left(e \cdot 2N/d\right)^d \leq (2N)^d$, where $d$ is the VC-dimension of the hypothesis set $\mathcal{H}$. Finally, by setting Eqn. 31 to $\delta$ and solve for the error tolerance $\epsilon$ will complete the proof. $\qquad \square$

Lemma A.2 asserts that altering or merging multiple target functions does not impact the generalization error term, as long as the sum of the weights for each loss $\sum_j \beta_j$ remains constant and the same dataset $\mathcal{X}$ is used for estimation. Next we highligt the Lemma 3 in [5] again, as it will be utilized for proving 3.3.

**Lemma A.3.** *For any hypothesis $h, h' \in \mathcal{H}$ and any two different domains $\mathcal{D}, \mathcal{D}'$,*

$$|\epsilon_{\mathcal{D}}(h, h') - \epsilon_{\mathcal{D}'}(h, h')| \leq \tfrac{1}{2} d_{\mathcal{H}\Delta\mathcal{H}}(\mathcal{D}, \mathcal{D}'). \quad (32)$$

*Proof.* By definition, we have

$$d_{\mathcal{H}\Delta\mathcal{H}}(\mathcal{D}, \mathcal{D}') = 2 \sup_{h, h' \in \mathcal{H}} |\mathbb{P}_{\boldsymbol{x} \sim \mathcal{D}}[h(\boldsymbol{x}) \neq h'(\boldsymbol{x})] - \mathbb{P}_{\boldsymbol{x} \sim \mathcal{D}'}[h(\boldsymbol{x}) \neq h'(\boldsymbol{x})]|$$

$$= 2 \sup_{h, h' \in \mathcal{H}} |\epsilon_{\mathcal{D}}(h, h') - \epsilon_{\mathcal{D}'}(h, h')|$$

$$\geq 2 |\epsilon_{\mathcal{D}}(h, h') - \epsilon_{\mathcal{D}'}(h, h')|. \qquad \square$$

Now we are ready to prove the main lemmas and theorems in the main body of our work.

**Lemma 3.1 (ERM-Based Generalization Bound).** *Let $\mathcal{H}$ be a hypothesis space of VC dimension $d$. When domain $t$ arrives, there are $N_t$ data points from domain $t$ and $\widetilde{N}_i$ data points from each previous domain $i < t$ in the memory bank. With probability at least $1 - \delta$, we have:*

$$\sum_{i=1}^{t} \epsilon_{\mathcal{D}_i}(h) \leq \sum_{i=1}^{t} \widehat{\epsilon}_{\mathcal{D}_i}(h) + \sqrt{\left(\frac{1}{N_t} + \sum_{i=1}^{t-1} \frac{1}{\widetilde{N}_i}\right)\left(8d \log\left(\tfrac{2eN}{d}\right) + 8 \log\left(\tfrac{2}{\delta}\right)\right)}. \quad (33)$$

*Proof.* Simply using Lemma A.1 and setting $\alpha_i = 1$ for every $i \in [t]$ completes the proof. $\qquad\square$

**Lemma 3.2** (**Intra-Domain Model-Based Bound**). *Let $h \in \mathcal{H}$ be an arbitrary function in the hypothesis space $\mathcal{H}$, and $H_{t-1}$ be the model trained after domain $t-1$. The domain-specific error $\epsilon_{\mathcal{D}_i}(h)$ on the previous domain $i$ has an upper bound:*

$$\epsilon_{\mathcal{D}_i}(h) \leq \epsilon_{\mathcal{D}_i}(h, H_{t-1}) + \epsilon_{\mathcal{D}_i}(H_{t-1}), \tag{34}$$

*where $\epsilon_{\mathcal{D}_i}(h, H_{t-1}) \triangleq \mathbb{E}_{\boldsymbol{x} \sim \mathcal{D}_i}[h(\boldsymbol{x}) \neq H_{t-1}(\boldsymbol{x})]$.*

*Proof.* By applying the triangle inequality [5] of the 0-1 loss function, we have

$$\begin{aligned}
\epsilon_{\mathcal{D}_i}(h) &= \epsilon_{\mathcal{D}_i}(h, f_i) \\
&\leq \epsilon_{\mathcal{D}_i}(h, H_{t-1}) + \epsilon_{\mathcal{D}_i}(H_{t-1}, f_i) \\
&= \epsilon_{\mathcal{D}_i}(h, H_{t-1}) + \epsilon_{\mathcal{D}_i}(H_{t-1}). \qquad\square
\end{aligned}$$

**Lemma 3.3** (**Cross-Domain Model-Based Bound**). *Let $h \in \mathcal{H}$ be an arbitrary function in the hypothesis space $\mathcal{H}$, and $H_{t-1}$ be the function trained after domain $t-1$. The domain-specific error $\epsilon_{\mathcal{D}_i}(h)$ evaluated on the previous domain $i$ then has an upper bound:*

$$\epsilon_{\mathcal{D}_i}(h) \leq \epsilon_{\mathcal{D}_t}(h, H_{t-1}) + \tfrac{1}{2}d_{\mathcal{H}\Delta\mathcal{H}}(\mathcal{D}_i, \mathcal{D}_t) + \epsilon_{\mathcal{D}_i}(H_{t-1}), \tag{35}$$

*where $d_{\mathcal{H}\Delta\mathcal{H}}(\mathcal{P}, \mathcal{Q}) = 2\sup_{h \in \mathcal{H}\Delta\mathcal{H}} |\Pr_{x \sim \mathcal{P}}[h(x) = 1] - \Pr_{x \sim \mathcal{Q}}[h(x) = 1]|$ denotes the $\mathcal{H}\Delta\mathcal{H}$-divergence between distribution $\mathcal{P}$ and $\mathcal{Q}$, and $\epsilon_{\mathcal{D}_t}(h, H_{t-1}) \triangleq \mathbb{E}_{\boldsymbol{x} \sim \mathcal{D}_t}[h(\boldsymbol{x}) \neq H_{t-1}(\boldsymbol{x})]$.*

*Proof.* By the triangle inequality used above and Lemma A.3, we have

$$\begin{aligned}
\epsilon_{\mathcal{D}_i}(h) &\leq \epsilon_{\mathcal{D}_i}(h, H_{t-1}) + \epsilon_{\mathcal{D}_i}(H_{t-1}) \\
&= \epsilon_{\mathcal{D}_i}(h, H_{t-1}) + \epsilon_{\mathcal{D}_i}(H_{t-1}) - \epsilon_{\mathcal{D}_t}(h, H_{t-1}) + \epsilon_{\mathcal{D}_t}(h, H_{t-1}) \\
&\leq \epsilon_{\mathcal{D}_i}(H_{t-1}) + |\epsilon_{\mathcal{D}_i}(h, H_{t-1}) - \epsilon_{\mathcal{D}_t}(h, H_{t-1})| + \epsilon_{\mathcal{D}_t}(h, H_{t-1}) \\
&\leq \epsilon_{\mathcal{D}_t}(h, H_{t-1}) + \tfrac{1}{2}d_{\mathcal{H}\Delta\mathcal{H}}(\mathcal{D}_i, \mathcal{D}_t) + \epsilon_{\mathcal{D}_i}(H_{t-1}). \qquad\square
\end{aligned}$$

**Theorem 3.4** (**Unified Generalization Bound for All Domains**). *Let $\mathcal{H}$ be a hypothesis space of VC dimension $d$. Let $N = N_t + \sum_i^{t-1} \widetilde{N}_i$ denoting the total number of data points available to the training of current domain $t$, where $N_t$ and $\widetilde{N}_i$ denote the numbers of data points collected at domain $t$ and data points from the previous domain $i$ in the memory bank, respectively. With probability at least $1 - \delta$, we have:*

$$\begin{aligned}
\sum_{i=1}^{t} \epsilon_{\mathcal{D}_i}(h) \leq{} &\left\{ \sum_{i=1}^{t-1} [\gamma_i \widehat{\epsilon}_{\mathcal{D}_i}(h) + \alpha_i \widehat{\epsilon}_{\mathcal{D}_i}(h, H_{t-1})] \right\} + \left\{ \widehat{\epsilon}_{\mathcal{D}_t}(h) + (\sum_{i=1}^{t-1} \beta_i)\widehat{\epsilon}_{\mathcal{D}_t}(h, H_{t-1}) \right\} \\
&+ \tfrac{1}{2}\sum_{i=1}^{t-1} \beta_i d_{\mathcal{H}\Delta\mathcal{H}}(\mathcal{D}_i, \mathcal{D}_t) + \sum_{i=1}^{t-1}(\alpha_i + \beta_i)\epsilon_{\mathcal{D}_i}(H_{t-1}) \\
&+ \sqrt{\left( \frac{(1+\sum_{i=1}^{t-1}\beta_i)^2}{N_t} + \sum_{i=1}^{t-1} \frac{(\gamma_i + \alpha_i)^2}{\widetilde{N}_i} \right) \left( 8d\log\left(\frac{2eN}{d}\right) + 8\log\left(\frac{2}{\delta}\right) \right)} \\
&\triangleq g(h, H_{t-1}, \Omega), \tag{36}
\end{aligned}$$

*where $\widehat{\epsilon}_{\mathcal{D}_i}(h, H_{t-1}) = \frac{1}{\widetilde{N}_i} \sum_{\boldsymbol{x} \in \widetilde{\mathcal{X}}_i} \mathbb{1}_{h(\boldsymbol{x}) \neq H_{t-1}(\boldsymbol{x})}$, $\widehat{\epsilon}_{\mathcal{D}_t}(h, H_{t-1}) = \frac{1}{N_t} \sum_{\boldsymbol{x} \in \mathcal{X}_i} \mathbb{1}_{h(\boldsymbol{x}) \neq H_{t-1}(\boldsymbol{x})}$, and $\Omega \triangleq \{\alpha_i, \beta_i, \gamma_i\}_{i=1}^{t-1}$.*

*Proof.* By applying Lemma 3.2 and Lemma 3.3 to each of the past domains, we have

$$\begin{aligned}
\epsilon_{\mathcal{D}_i}(h) &= (\alpha_i + \beta_i + \gamma_i)\epsilon_{\mathcal{D}_i}(h) \\
&\leq \gamma_i \epsilon_{\mathcal{D}_i}(h) + \alpha_i[\epsilon_{\mathcal{D}_i}(h, H_{t-1}) + \epsilon_{\mathcal{D}_i}(H_{t-1})] \\
&\quad + \beta_i[\epsilon_{\mathcal{D}_i}(h, H_{t-1}) + \epsilon_{\mathcal{D}_t}(h, H_{t-1}) + \tfrac{1}{2}d_{\mathcal{H}\Delta\mathcal{H}}(\mathcal{D}_i, \mathcal{D}_t)].
\end{aligned}$$

Table 4: Unification of existing methods under UDIL, when certain conditions are achieved.

| | $\alpha_i$ | $\beta_i$ | $\gamma_i$ | Transformed Objective | Condition |
|---|---|---|---|---|---|
| UDIL (Ours) | $[0,1]$ | $[0,1]$ | $[0,1]$ | - | - |
| LwF [52] | 0 | 1 | 0 | $\mathcal{L}_{\text{LwF}}(h) = \widehat{\ell}_{\mathcal{X}_t}(h) + \lambda_o\widehat{\ell}_{\mathcal{X}_t}(h, H_{t-1})$ | $\lambda_o = t-1$ |
| ER [75] | 0 | 0 | 1 | $\mathcal{L}_{\text{ER}}(h) = \widehat{\ell}_{B_t}(h) + \sum_{i=1}^{t-1}\frac{|B'_t|/(t-1)}{|B_t|}\widehat{\ell}_{B'_i}(h)$ | $|B_t| = \frac{|B'_t|}{(t-1)}$ |
| DER++ [8] | 1/2 | 0 | 1/2 | $\mathcal{L}_{\text{DER++}}(h) = \widehat{\ell}_{B_t}(h) + \frac{1}{2}\sum_{i=1}^{t-1}\frac{|B'_t|/(t-1)}{|B_t|}[\widehat{\ell}_{B'_i}(h) + \widehat{\ell}_{B'_i}(h, H_{t-1})]$ | $|B_t| = \frac{|B'_t|}{(t-1)}$ |
| iCaRL [74] | 1 | 0 | 0 | $\mathcal{L}_{\text{iCaRL}}(h) = \widehat{\ell}_{B_t}(h) + \sum_{i=1}^{t-1}\frac{|B'_t|/(t-1)}{|B_t|}\widehat{\ell}_{B'_i}(h, H_{t-1})$ | $|B_t| = \frac{|B'_t|}{(t-1)}$ |
| CLS-ER [4] | $\frac{\lambda}{\lambda+1}$ | 0 | $\frac{1}{\lambda+1}$ | $\mathcal{L}_{\text{CLS-ER}}(h) = \widehat{\ell}_{B_t}(h) + \sum_{i=1}^{t-1}\frac{1}{t-1}\widehat{\ell}_{B'_i}(h) + \sum_{i=1}^{t-1}\frac{\lambda}{t-1}\widehat{\ell}_{B'_i}(h, H_{t-1})$ | $\lambda = t-2$ |
| ESM-ER [80] | $\frac{\lambda}{\lambda+1}$ | 0 | $\frac{1}{\lambda+1}$ | $\mathcal{L}_{\text{ESM-ER}}(h) = \widehat{\ell}_{B_t}(h) + \sum_{i=1}^{t-1}\frac{1}{r(t-1)}\widehat{\ell}_{B'_i}(h) + \sum_{i=1}^{t-1}\frac{\lambda}{r(t-1)}\widehat{\ell}_{B'_i}(h, H_{t-1})$ | $\begin{cases}\lambda = -1 + r(t-1)\\ r = 1 - e^{-1}\end{cases}$ |
| BiC [100] | $\frac{t-1}{2t-1}$ | $\frac{t-1}{2t-1}$ | $\frac{1}{2t-1}$ | $\mathcal{L}_{\text{BiC}}(h) = \widehat{\ell}_{B_t}(h) + \sum_{i=1}^{t-1}\frac{(t-1)|B_i|}{|B_t|}\widehat{\ell}_{B'_i}(h, H_{t-1})$ $+ (t-1)\widehat{\ell}_{B_t}(h, H_{t-1}) + \sum_{i=1}^{t-1}\frac{|B_i|}{|B_t|}\widehat{\ell}_{B'_i}(h)$ | $|B_i| = |B_t|$ |

Re-organizing the terms will give us

$$\sum_{i=1}^{t}\epsilon_{\mathcal{D}_i}(h) \leq \left\{\sum_{i=1}^{t-1}[\gamma_i\epsilon_{\mathcal{D}_i}(h) + \alpha_i\epsilon_{\mathcal{D}_i}(h, H_{t-1})]\right\} + \left\{\epsilon_{\mathcal{D}_t}(h) + (\sum_{i=1}^{t-1}\beta_i)\epsilon_{\mathcal{D}_t}(h, H_{t-1})\right\}$$
$$+ \frac{1}{2}\sum_{i=1}^{t-1}\beta_i d_{\mathcal{H}\Delta\mathcal{H}}(\mathcal{D}_i, \mathcal{D}_t) + \sum_{i=1}^{t-1}(\alpha_i + \beta_i)\epsilon_{\mathcal{D}_i}(H_{t-1}). \tag{37}$$

Then applying Lemma A.1 and Lemma A.2 jointly to Eqn. 37 will complete the proof. $\square$

# B   UDIL as a Unified Framework

In this section, we will delve into a comprehensive discussion of our UDIL framework, which serves as a unification of numerous existing methods. It is important to note that we incorporate methods designed for task incremental and class incremental scenarios that can be easily adapted to our domain incremental learning. To provide clarity, we will present the corresponding coefficients $\{\alpha_i, \beta_i, \gamma_i\}$ of each method within our UDIL framework (refer to Table 4). Furthermore, we will explore the conditions under which these coefficients are included in this unification process.

**Learning without Forgetting (LwF) [52]** was initially proposed for task-incremental learning, incorporating a combination of *shared parameters* and *task-specific parameters*. This framework can be readily extended to domain incremental learning by setting all "domain-specific" parameters to be the same in a static model architecture. LwF was designed for the strict continual learning setting, where no data from past tasks is accessible. To overcome this limitation, LwF records the predictions of the history model $H_{t-1}$ on the current data $\mathcal{X}_t$ at the beginning of the new task $t$. Subsequently, knowledge distillation (as defined in Definition 4.2) is performed to mitigate forgetting:

$$\mathcal{L}_{\text{old}}(h, H_{t-1}) \triangleq -\frac{1}{N_t}\sum_{\boldsymbol{x}\in\mathcal{X}_t}\sum_{k=1}^{K}[H_{t-1}(\boldsymbol{x})]_k \cdot [\log([h(\boldsymbol{x})]_k)] = \widehat{\ell}_{\mathcal{X}_t}(h, H_{t-1}), \tag{38}$$

where $H_{t-1}(\boldsymbol{x}), h(\boldsymbol{x}) \in \mathbb{R}^K$ are the class distribution of $\boldsymbol{x}$ over $K$ classes produced by the history model and current model, respectively. The loss for learning the current task $\mathcal{L}_{\text{new}}$ is defined as

$$\mathcal{L}_{\text{new}}(h) \triangleq -\frac{1}{N_t}\sum_{(\boldsymbol{x},y)\in\mathcal{S}_t}\sum_{k=1}^{K}\mathbb{1}_{y=k} \cdot [\log([h(\boldsymbol{x})]_k)] = \widehat{\ell}_{\mathcal{X}_t}(h). \tag{39}$$

LwF uses a "loss balance weight" $\lambda_o$ to balance two losses, which gives us its final loss for training:

$$\mathcal{L}_{\text{LwF}}(h) \triangleq \mathcal{L}_{\text{new}}(h) + \lambda_o \cdot \mathcal{L}_{\text{old}}(h, H_{t-1}). \tag{40}$$

In LwF, the default setting assumes the presence of two domains (tasks) with $\lambda_o = 1$. However, it is possible to learn multiple domains continuously using LwF's default configuration. To achieve this, the current domain $t$ can be weighed against the number of previous domains (1 versus $t-1$). Specifically, if there is no preference for any particular domain, $\lambda_o$ should be set to $t-1$. Remarkably, this is equivalent to setting $\{\beta_i = 1, \alpha_i = \gamma_i = 0\}$ in our UDIL framework (Row 2 in Table 4).

**Experience Replay (ER) [75]** serves as the fundamental operation for replay-based continual learning methods. It involves storing and replaying a subset of examples from past domains during training. Following the description and implementation provided by [8], ER operates as follows: during each training iteration on domain $t$, a mini-batch $B_t$ of examples is sampled from the current domain, along with a mini-batch $B_t'$ from the memory. These two mini-batches are then concatenated into a larger mini-batch $(B_t \cup B_t')$, upon which average gradient descent is performed:

$$\mathcal{L}_{\text{ER}}(h) = \widehat{\ell}_{B_t \cup B_t'}(h) \tag{41}$$

$$= \tfrac{1}{|B_t|+|B_t'|} \sum_{(\boldsymbol{x},y) \in B_t \cup B_t'} \sum_{k=1}^{K} \mathbb{1}_{y=k} \cdot [\log([h(\boldsymbol{x})]_k)] \tag{42}$$

$$= \tfrac{|B_t|}{|B_t|+|B_t'|} \widehat{\ell}_{B_t}(h) + \tfrac{|B_t'|}{|B_t|+|B_t'|} \widehat{\ell}_{B_t'}(h). \tag{43}$$

Suppose that each time the mini-batch of past-domain data is perfectly balanced, meaning that each domain has the same number of examples in $B_t'$. In this case, Eqn. 43 can be further decomposed as follows:

$$\mathcal{L}_{\text{ER}}(h) = \tfrac{|B_t|}{|B_t|+|B_t'|} \widehat{\ell}_{B_t}(h) + \sum_{i=1}^{t-1} \tfrac{|B_t'|/(t-1)}{|B_t|+|B_t'|} \widehat{\ell}_{B_i'}(h), \tag{44}$$

where $B_i' = \{(\boldsymbol{x},y) | (\boldsymbol{x},y) \in (B_t' \cap M_i)\}$ is the subset of the mini-batch that belongs to domain $i$.

Now, by dividing both sides of Eqn. 44 by $(|B_t|+|B_t'|/|B_t|)$ and comparing it to Theorem 3.4, we can include ER in our UDIL framework when the condition $|B_t| = |B_t'|/(t-1)$ is satisfied. In this case, ER is equivalent to $\{\alpha_i = \beta_i = 0, \gamma_i = 1\}$ in UDIL (Row 3 in Table 4). It is important to note that this condition is not commonly met throughout the entire process of continual learning. It can be achieved by linearly scaling up the size of the mini-batch from the memory (which is feasible in the early domains) or by linearly scaling down the mini-batch from the current-domain data (which may cause a drop in model performance). It is worth mentioning that this incongruence *highlights the intrinsic bias of the original ER setting towards current domain learning* and cannot be rectified by adjusting the batch sizes of the current domain or the memory. However, it does not weaken our claim of unification.

**Dark Experience Replay (DER++) [8]** includes an additional dark experience replay, i.e., knowledge distillation on the past domain exemplars, compared to ER [75]. Now under the same assumptions (balanced sampling strategy and $|B_t| = |B_t'|/(t-1)$) as discussed for ER, we can utilize Eqn. 44 to transform the DER++ loss as follows:

$$\mathcal{L}_{\text{DER++}}(h) = \tfrac{|B_t|}{|B_t|+|B_t'|} \widehat{\ell}_{B_t}(h) + \tfrac{1}{2} \sum_{i=1}^{t-1} \tfrac{|B_t'|/(t-1)}{|B_t|+|B_t'|} \widehat{\ell}_{B_i'}(h) + \tfrac{1}{2} \sum_{i=1}^{t-1} \tfrac{|B_t'|/(t-1)}{|B_t|+|B_t'|} \widehat{\ell}_{B_i'}(h, H_{t-1}). \tag{45}$$

In this scenario, DER++ is equivalent to $\{\alpha_i = \gamma_i = 1/2, \beta_i = 0\}$ in UDIL (Row 4 in Table 4).

**Incremental Classifier and Representation Learning (iCaRL) [74]** was initially proposed for class incremental learning. It decouples learning the representations and final classifier into two individual phases. During training, iCaRL adopts an incrementally increasing linear classifier. Different from traditional design choice of multi-class classification where the softmax activation layer and multi-class cross entropy loss are used jointly, iCaRL models multi-class classification as multi-label classification [107, 86, 106]. Suppose there are $K$ classes in total, and we denote the one-hot label vector of the input $\boldsymbol{x}$ as $\boldsymbol{y} \in \mathbb{R}^K$ where $y_j \triangleq \mathbb{1}_{f(\boldsymbol{x})=j}$. Then the *multi-label learning objective* treats each dimension of the output logits as a score for binary classification, which is computed as follows:

$$\widehat{\ell}_{\mathcal{X}}(h) \triangleq \tfrac{-1}{|\mathcal{X}|} \sum_{\boldsymbol{x} \in \mathcal{X}} \sum_{j=1}^{K} \left[ y_j \log\left([h(\boldsymbol{x})]_j\right) + (1-y_j) \log\left(1 - [h(\boldsymbol{x})]_j\right) \right]. \tag{46}$$

When a new task that contains $K'$ new classes is presented, iCaRL adds additional $K'$ new dimensions to the final linear classifier.

In iCaRL training, the new classes and the old classes are treated differently. For new classes, it trains the representations of the network with the ground-truth labels ($f(\boldsymbol{x})$ in the original paper), and for old classes, it uses the history model's output ($H_{t-1}(\boldsymbol{x})$) as the learning target (i.e., pseudo labels). Each data point is treated with the same level of importance during iCaRL's training procedure. Hence after translating the loss function of iCaRL in the context of class-incremental learning to domain-incremental learning, we have

$$\mathcal{L}_{\text{iCaRL}}(h) = N_t \cdot \widehat{\ell}_{\mathcal{X}_t}(h) + \sum_{i=1}^{t-1} \widetilde{N}_i \cdot \widehat{\ell}_{\mathcal{X}_i}(h, H_{t-1}), \tag{47}$$

where in $\widehat{\ell}_{\mathcal{X}_i}(h, H_{t-1})$, we follow the same definition as in the distillation loss in Eqn. 4.2 and replace the ground-truth label with the soft label for iCaRL.

Similar to ER [75] and DER++ [8] as analyzed before, the equation above does not naturally fall into the realm of unification provided in UDIL. To achieve this, we need to sample the data from the current domain and exemplars in the memory independently, i.e., assuming $B_t$ and $B_i$. By applying the same rule of $\alpha_i + \beta_i + \gamma_i = 1$, we now have that for iCaRL, the corresponding coefficients are $\{\alpha_i = {}^{B_i}/_{B_t} = 1, \beta_i = \gamma_i = 0\}$ (Row 5 in Table 4).

**Complementary Learning System based Experience Replay (CLS-ER) [4]** involves the mainte-nance of two history models, namely the plastic model $H_{t-1}^{(p)}$ and the stable model $H_{t-1}^{(s)}$, throughout the continual training process of the working model $h$. Following each update of the working model, the two history models are stochastically updated at different rates using exponential moving averages (EMA) of the working model's *parameters*:

$$H_{t-1}^{(i)} \leftarrow \alpha^{(i)} \cdot H_{t-1}^{(i)} + (1 - \alpha^{(i)}) \cdot h, \qquad i \in \{p, s\}, \tag{48}$$

where $\alpha^{(p)} \leq \alpha^{(s)}$ is set such that the plastic model undergoes rapid updates, allowing it to swiftly adapt to newly acquired knowledge, while the stable model maintains a "long-term memory" spanning multiple tasks. Throughout training, CLS-ER assesses the certainty generated by both history models and employs the logits from the more certain model as the target for knowledge distillation.

In the general formulation of the UDIL framework, the history model $H_{t-1}$ is not required to be a single model with the same architecture as the current model $h$. In fact, if there are no constraints on memory consumption, we have the flexibility to train and preserve a domain-specific model $H_i$ for each domain $i$. During testing, we can simply select the prediction with the highest certainty from each domain-specific model. From this perspective, the "two-history-model system" employed in CLS-ER can be viewed as a specific and limited version of the all-domain history models. Consequently, we can combine the two models used in CLS-ER into a single history model $H_{t-1}$ as follows:

$$H_{t-1}(\boldsymbol{x}) \triangleq \begin{cases} H_{t-1}^{(p)}(\boldsymbol{x}) & \text{if } [H_{t-1}^{(p)}(\boldsymbol{x})]_y > [H_{t-1}^{(s)}(\boldsymbol{x})]_y \\ H_{t-1}^{(s)}(\boldsymbol{x}) & \text{o.w.} \end{cases} \tag{49}$$

where $(\boldsymbol{x}, y) \in \mathcal{M}$ is an arbitrary exemplar stored in the memory bank.

At each iteration of training, CLS-ER samples a mini-batch $B_t$ from the current domain and a mini-batch $B_t'$ from the episodic memory. It then concatenates $B_t$ and $B_t'$ for the cross entropy loss minimization with the ground-truth labels, and uses $B_t'$ to minimize the MSE loss between the logits of $h$ and $H_{t-1}$. To align the loss formulation of CLS-ER with that of ESM-ER [80], here we consider the scenarios where the losses evaluated on $B_t$ and $B_t'$ are individually calculated, i.e., we consider $\widehat{\ell}_{B_t}(h) + \widehat{\ell}_{B_t'}(h)$ instead of $\widehat{\ell}_{B_t \cup B_t'}(h)$. Based on the assumption from [34], the MSE loss on the logits is equivalent to the cross-entropy loss on the predictions under certain conditions. Therefore, following the same balanced sampling strategy assumptions as in ER, the original CLS-ER training objective can be transformed as follows:

$$\mathcal{L}_{\text{CLS-ER}}(h) = \widehat{\ell}_{B_t}(h) + \widehat{\ell}_{B_t'}(h) + \lambda \widehat{\ell}_{B_t'}(h, H_{t-1}) \tag{50}$$

$$= \widehat{\ell}_{B_t}(h) + \sum_{i=1}^{t-1} \frac{1}{t-1} \widehat{\ell}_{B_i'}(h) + \sum_{i=1}^{t-1} \frac{\lambda}{t-1} \widehat{\ell}_{B_i'}(h, H_{t-1}). \tag{51}$$

Therefore, by imposing the constraint $\alpha_i + \beta_i + \gamma_i = 1$, we find that $\lambda = t - 2$. Substituting this value back into $\lambda/t-1$ yields the equivalence that CLS-ER corresponds to $\{\alpha_i = \lambda/\lambda+1, \beta_i = 0, \gamma_i = 1/\lambda+1\}$ in UDIL, where $\lambda = t - 2$ (Row 6 in Table 4).

**Error Sensitivity Modulation based Experience Replay (ESM-ER)** [80] builds upon CLS-ER by incorporating an additional error sensitivity modulation module. The primary goal of ESM-ER is to mitigate sudden representation drift caused by excessively large loss values during current-domain learning. Let's consider $(\boldsymbol{x}, y) \sim \mathcal{D}_t$, which represents a sample from the current domain batch. In ESM-ER, the cross-entropy loss value of this sample is evaluated using the stable model $H_{t-1}^{(s)}$ and can be expressed as:

$$\ell(\boldsymbol{x}, y) = -\log([H_{t-1}^{(s)}(\boldsymbol{x})]_y). \tag{52}$$

To screen out those samples with a high loss value, ESM-ER assigns each sample a weight $\lambda$ by comparing the loss with their expectation value, for which ESM-ER uses a running estimate $\mu$ as its replacement. This can be formulated as follows:

$$\lambda(\boldsymbol{x}) = \begin{cases} 1 & \text{if } \ell(\boldsymbol{x}, y) \leq \beta \cdot \mu \\ \frac{\mu}{\ell(\boldsymbol{x}, y)} & \text{o.w.} \end{cases} \tag{53}$$

where $\beta$ is a hyperparameter that determines the margin for a sample to be classified as low-loss. For the sake of analysis, we make the following assumptions: (i) $\beta = 1$; (ii) the actual expected loss value $\mathbb{E}_{\boldsymbol{x}, y}[\ell(\boldsymbol{x}, y)]$ is used instead of the running estimate $\mu$; (iii) a *hard screening mechanism* is employed instead of the current re-scaling approach. Based on these assumptions, we determine the sample-wise weights $\lambda^\star$ according to the following rule:

$$\lambda^\star(\boldsymbol{x}) = \begin{cases} 1 & \text{if } \ell(\boldsymbol{x}, y) \leq \mathbb{E}_{\boldsymbol{x}, y}[\ell(\boldsymbol{x}, y)] \\ 0 & \text{o.w.} \end{cases} \tag{54}$$

Under the assumption that the loss value $\ell(\boldsymbol{x}, y)$ follows an exponential distribution, denoted as $\ell(\boldsymbol{x}, y) \sim \text{Exp}(\lambda_0)$, where the probability density function is given by $f(\ell(\boldsymbol{x}, y), \lambda_0) = \lambda_0 e^{-\lambda_0 \ell(\boldsymbol{x}, y)}$, we can calculate the expectation of the loss as $\mathbb{E}_{\boldsymbol{x}, y}[\ell(\boldsymbol{x}, y)] = 1/\lambda_0$. Based on this, we can now determine the expected ratio $r$ of the *unscreened samples* in a mini-batch using the following equation:

$$r = \int_0^{\frac{1}{\lambda_0}} 1 \cdot \lambda_0 e^{-\lambda_0 \ell(\boldsymbol{x}, y)} \, d\ell(\boldsymbol{x}, y) = \int_0^1 e^{-y} dy = (1 - e^{-1}). \tag{55}$$

The ratio $r$ represents the proportion of effective samples in the current-domain batch, as the weights $\lambda^\star(\boldsymbol{x})$ of the remaining samples are set to 0 due to their high loss value. Consequently, the original training loss of ESM-ER can be transformed as follows:

$$\mathcal{L}_{\text{ESM-ER}}(h) = r \cdot \widehat{\ell}_{B_t}(h) + \widehat{\ell}_{B'_t}(h) + \lambda \widehat{\ell}_{B'_t}(h, H_{t-1}) \tag{56}$$

$$= r \cdot \widehat{\ell}_{B_t}(h) + \sum_{i=1}^{t-1} \frac{1}{t-1} \widehat{\ell}_{B'_i}(h) + \sum_{i=1}^{t-1} \frac{\lambda}{t-1} \widehat{\ell}_{B'_i}(h, H_{t-1}). \tag{57}$$

After applying the constraint of $\alpha_i + \beta_i + \gamma_i = 1$, we obtain $\lambda = r \cdot (t - 1) - 1$. Substituting this value back into $\lambda/r(t-1)$, we find that ESM-ER is equivalent to $\{\alpha_i = \lambda/\lambda+1, \beta_i = 0, \gamma_i = 1/\lambda+1\}$ in UDIL, where $\lambda = r \cdot (t - 1) - 1 = (1 - e^{-1})(t - 1) - 1$ should be set (Row 7 in Table 4).

**Bias Correction (BiC)** [100] was the first to apply class incremental learning at a large scale, covering extensive image classification datasets including ImageNet (1,000 classes, [76]) and MS-Celeb-1M (10,000 classes, [31]). Similar to iCaRL [74], BiC treats each data example (from new classes' data and the memory) with the same level of importance. Different from its previous work, the distillation loss in BiC ($L_d$ in the original paper) implicitly includes the cross-domain distillation loss described by Lemma 3.3, as it computes the old classifier's output evaluated on the new data (considering only old classes). In addition, BiC further re-balances the classification loss ($L_c$ in the original paper) and the distillation loss ($L_d$) based on the number of old classes $n$ and new classes $m$ as follows:

$$\mathcal{L}_{\text{BiC}} = \frac{n}{n+m} \cdot L_d + \frac{m}{n+m} \cdot L_c. \tag{58}$$

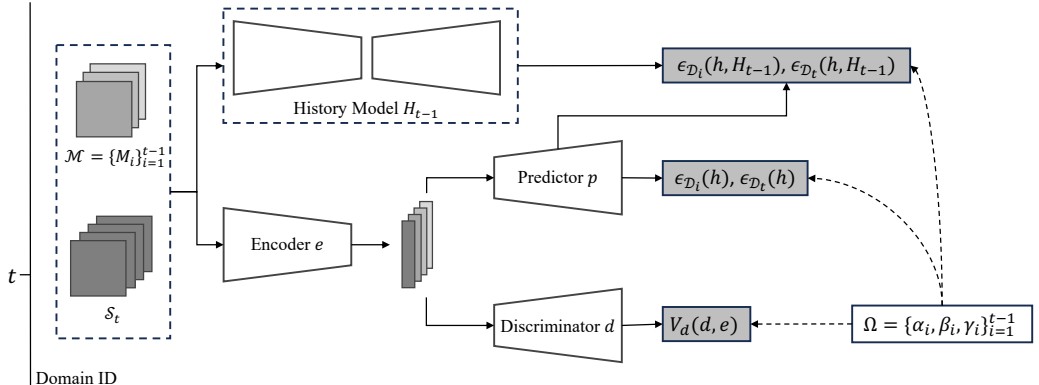

Figure 2: **Diagram of UDIL.** $\mathcal{M} = \{M_i\}_{i=1}^{t-1}$ represents the memory bank that stores all the past exemplars. $\mathcal{S}_t$ corresponds to the dataset from the current domain $t$, and the current model $h = p \circ e$ is depicted separately in the diagram. The three different categories of losses are illustrated in the dark rectangles, while the weighting effect of the learned replay coefficient $\Omega = \{\alpha_i, \beta_i, \gamma_i\}_{i=1}^{t-1}$ is depicted using dashed lines.

By interpreting the distillation loss $L_d$ as the combination of the intra-domain loss (Lemma 3.2) and the cross-domain distillation loss (Lemma 3.3), and substituting $(n, m)$ with $(t-1, 1)$ due to the consistent number of classes in each domain, we arrive at the BiC model for DIL:

$$\mathcal{L}_{\text{BiC}}(h) = \frac{t-1}{t} \left[ N_t \cdot \widehat{\ell}_{\mathcal{X}_t}(h, H_{t-1}) + \sum_{i=1}^{t-1} \widetilde{N}_i \cdot \widehat{\ell}_{\mathcal{X}_i}(h, H_{t-1}) \right]$$

$$+ \frac{1}{t} \left[ N_t \cdot \widehat{\ell}_{\mathcal{X}_t}(h) + \sum_{i=1}^{t-1} \widetilde{N}_i \cdot \widehat{\ell}_{\mathcal{X}_i}(h) \right]. \tag{59}$$

The equation above relies on the number of the current-domain examples and the exemplars (i.e., memory), which is not constant in practice. Hence we replace $(N_t, \widetilde{N}_i)$ with the mini-batch size $(|B_t|, |B_i|)$. After re-organizing the equation above, we have

$$\mathcal{L}_{\text{BiC}}(h) = \widehat{\ell}_{B_t}(h) + \sum_{i=1}^{t-1} \frac{(t-1)|B_i|}{|B_t|} \cdot \widehat{\ell}_{B'_i}(h, H_{t-1})$$

$$+ (t-1) \cdot \widehat{\ell}_{B_t}(h, H_{t-1}) + \sum_{i=1}^{t-1} \frac{|B_i|}{|B_t|} \cdot \widehat{\ell}_{B'_i}(h). \tag{60}$$

The immediate observatio is that *BiC exhibits a significant bias towards retaining knowledge from past domains*. This is evident in the coefficient of coefficient of cross-domain distillation summed over the past domains $\sum_{i=1}^{t-1} \beta_i = (t-1)$, which violates the constraints posed in this work ($\alpha_i + \beta_i + \gamma_i = 1$). However, if we slightly relax BiC's formulation and focus on the relative ratios of the three coefficient, we get $\alpha_i : \beta_i : \gamma_i = (t-1)B_i : (t-1)B_t : B_i$. Further applying the constraint $\alpha_i + \beta_i + \gamma_i = 1$ and assuming $B_i = B_t$ to enforce a balanced sampling strategy over different domains, we arrive at $\{\alpha_i = \beta_i = {}^{t-1}/_{2t-1}, \gamma_i = {}^{1}/_{2t-1}\}$ (Row 8 in Table 4).

## C  Implementation Details of UDIL

This section delves into the implementation details of the UDIL algorithm. The algorithmic description of UDIL is presented in Algorithm 1 and a diagram is presented in Fig. 2. However, there are several practical issues to be further addressed here, including (i) how to exert the constraints of probability simplex ($[\alpha_i, \beta_i, \gamma_i] \in \mathbb{S}^2$) and (ii) how the memory is maintained. These two problems will be addressed in Sec. C.1 and Sec. C.2. Then, Sec. C.3 will introduce two auxiliary losses that improve the stability and domain alignment for the encoder during training. Next, Sec. C.4

will cover the evaluation metrics used in this paper. Finally, Sec. C.5 and Sec. C.6 will present a detailed introduction to the main baselines and the specific training schemes we follow for empirical evaluation.

## C.1  Modeling the Replay Coefficients $\Omega = \{\alpha_i, \beta_i, \gamma_i\}$

Instead of directly modeling $\Omega$ in a way such that it can be updated by gradient descent and satisfies the constraints that $\alpha_i + \beta_i + \gamma_i = 1$ and $\alpha_i, \beta_i, \gamma_i \geq 0$ at the same time, we use a set of logit variables $\{\bar{\alpha}_i, \bar{\beta}_i, \bar{\gamma}_i\} \in \mathbb{R}^3$ and the *softmax* function to indirectly calculate $\Omega$ during training. Concretely, we have:

$$
\begin{bmatrix} \alpha_i \\ \beta_i \\ \gamma_i \end{bmatrix} = \mathrm{softmax}\left( \begin{bmatrix} \bar{\alpha}_i \\ \bar{\beta}_i \\ \bar{\gamma}_i \end{bmatrix} \right) = \begin{bmatrix} \exp(\bar{\alpha}_i)/Z_i \\ \exp(\bar{\beta}_i)/Z_i \\ \exp(\bar{\gamma}_i)/Z_i \end{bmatrix}, \tag{61}
$$

where $Z_i = \exp(\bar{\alpha}_i) + \exp(\bar{\beta}_i) + \exp(\bar{\gamma}_i)$ is the normalizing constant. At the beginning of training on domain $t$, the logit variables $\{\bar{\alpha}_i, \bar{\beta}_i, \bar{\gamma}_i\} = \{0, 0, 0\}$ are initialized to all zeros, since we do not have any bias towards any upper bound. During training, they are updated in the same way as the other parameters with gradient descent.

## C.2  Memory Maintenance with Balanced Sampling

Different from DER++ [8] and its following work [4, 80] that use reservoir sampling [93] to maintain the episodic memory, UDIL adopts a random balanced sampling after training on each domain. To be more concrete, given a memory bank with fixed size $|\mathcal{M}|$, after domain $t$'s training is complete, we will assign each domain a quota of $|\mathcal{M}|/t$. For the current domain $t$, we will randomly sample $\lfloor |\mathcal{M}|/t \rfloor$ exemplars from its dataset; for all the previous domains $i \in [t-1]$, we will randomly swap out $\lceil |\mathcal{M}|/t-1 - |\mathcal{M}|/t \rceil$ exemplars from the memory to make sure each domain has roughly the same number of exemplars. To ensure a fair comparison, we use the same random balanced sampling strategy for all the other baselines. The following Algorithm 2 shows the detailed procedure of random balanced sampling.

---

**Algorithm 2** Balanced Sampling for UDIL

**Require:** memory bank $\mathcal{M} = \{M_i\}_{i=1}^{t-1}$, current domain dataset $\mathcal{S}_t$, domain ID $t$.

1: **for** $i = 1, \cdots, t-1$ **do**
2:      **for** $j = 1, \cdots, \lceil |\mathcal{M}|/t-1 - |\mathcal{M}|/t \rceil$ **do**
3:          $(\boldsymbol{x}, y) \leftarrow \mathrm{RandomSample}(M_i)$
4:          $(\boldsymbol{x}', y') \leftarrow \mathrm{RandomSample}(\mathcal{S}_t)$
5:          Swap $(\boldsymbol{x}', y')$ into $\mathcal{M}$, replacing $(\boldsymbol{x}, y)$
6:      **end for**
7: **end for**
8: **return** $\mathcal{M}$

---

## C.3  Improving Stability and Domain Alignment for the Embedding Distribution

In this section, we will examine the training loss employed to align the embedding distribution across distinct domains. As discussed in Sec. 4.2, we decompose the model $h$ into an encoder $e$ and a predictor $p$ as $h = p \circ e$. In order to establish a tighter upper bound proposed in Theorem 3.4, we introduce a discriminator $d$ that aims to distinguish embeddings based on domain IDs. Specifically, during the training process of UDIL, given a set of coefficients $\Omega = \{\alpha_i, \beta_i, \gamma_i\}$, both the encoder $e$ and the discriminator $d$ engage in the following minimax game:

$$
\min_e \max_d - \lambda_d V_d(d, e, \mathring{\Omega}), \tag{62}
$$

where $(\mathring{\cdot})$ represents the "copying-weights-and-stopping-gradients" operation, and $\lambda_d$ is a hyperparameter introduced to control the strength of domain embedding alignment. More specifically, the

value function of the mini-max game $V_d$ is defined as follows:

$$V_d(d, e, \overset{\circ}{\Omega}) = \left( \sum_{i=1}^{t-1} \overset{\circ}{\beta}_i \right) \frac{1}{N_i} \sum_{\boldsymbol{x} \in \mathcal{S}_t} \left[ -\log\left( [d(e(\boldsymbol{x}))]_t \right) \right] + \sum_{i=1}^{t-1} \frac{\overset{\circ}{\beta}_i}{\widetilde{N}_i} \sum_{\boldsymbol{x} \in \widetilde{\mathcal{X}}_i} \left[ -\log\left( [d(e(\boldsymbol{x}))]_i \right) \right]. \quad (63)$$

As previously mentioned, the practical effect of Eqn. 62 is to align the embedding distribution of different domains, thus enhancing the model's generalization ability to both previously encountered domains and those that may be encountered in the future.

However, actively altering the embedding space can lead to the well-known stability-plasticity dilemma [43, 96, 20, 39, 61]. This dilemma arises when the model needs to modify a significant number of parameters to achieve domain alignment for a new domain, potentially resulting in a mismatch between the predictor $p$ and the encoder $e$, which, in turn, can lead to catastrophic forgetting. Furthermore, it is worth noting that the adversarial training scheme described in Eqn. 62 is primarily designed for unsupervised domain alignment [108, 57, 26, 109, 17, 102, 101, 94], where the semantic labels in the target domain(s) are not available during training. This implies that the current adversarial training technique does not fully leverage the label information in domain incremental learning.

To tackle the aforementioned challenges, i.e., (i) maintaining a stable embedding distribution across all domains and (ii) accelerating domain alignment with label information, we incorporate two additional auxiliary losses: (i) the *past embedding distillation loss* [40, 65] and (ii) the *supervised contrastive loss* [42, 29]. It's important to note that these auxiliary losses, in conjunction with the adversarial feature alignment loss $V_d$, operate exclusively on the encoder space of the model and do not impact the original objectives outlined in Theorem 3.4, provided that encoder $e$ and predictor $p$ remain sufficiently strong. Therefore, the two losses are used to simply stabilize the training, without compromising the theoretical significance of our work.

In **past embedding distillation**, also known as *representation(al) distillation* or *embedding distillation* in previous work on continual learning [40, 65, 59], the model stores the embeddings of the memory after being trained on domain $t-1$. It then uses them to constrain the encoder's behavior: the features produced on these samples should not change too much during the current domain training. After decoupling the history model $H_{t-1} = P_{t-1} \circ E_{t-1}$, where $E_{t-1}$ is the history encoder and $P_{t-1}$ is history predictor, we define the past embedding distillation loss $V_p$ as

$$V_p(e) = \sum_{i=1}^{t-1} \mathbb{E}_{\boldsymbol{x} \sim \mathcal{D}_i} \left[ \| e(\boldsymbol{x}) - E_{t-1}(\boldsymbol{x}) \|_2^2 \right]. \quad (64)$$

The loss above promotes the stability of the embedding distribution from past domains. When combined with the adversarial embedding alignment loss in Eqn. 62, it encourages the embedding distribution of the current domain to match those of the previous domains, but not vice versa.

As supervised variations of contrastive learning [12, 32, 14, 83, 65, 13, 84, 85], the **supervised contrastive loss** [42, 29] will compensate for the fact that Eqn. 62 does not utilize the label information to align different domains, and therefore lack in the efficiency of aligning two domains' embedding distribution. The supervised contrastive learning pulls together the embeddings of the same label and pushes apart those with distinct labels. Notably, it is done in a "domain-agnostic" manner, i.e., the domain labels are not considered. Denoting as $\mathcal{P} = \frac{1}{t} \sum_i^t \mathcal{D}_i$ the combined data distribution of all domains and as $\mathcal{P}_{\cdot|y}$ the data distribution given the class label $y \in [K]$, the supervised contrastive loss for embedding alignment is defined as follows:

$$V_s(e) = \mathbb{E}_y \mathbb{E}_{\substack{(\boldsymbol{x}_1, \boldsymbol{x}_2) \sim \mathcal{P}_{\cdot|y} \\ \{\boldsymbol{u}_i\}_{i=1}^B \sim \mathcal{P}}} \left[ -\log \frac{\exp\{-s_e(\boldsymbol{x}_1, \boldsymbol{x}_2)\}}{\exp\{-s_e(\boldsymbol{x}_1, \boldsymbol{x}_2)\} + \sum_{i=1}^B \exp\{-s_e(\boldsymbol{x}_1, \boldsymbol{u}_i)\}} \right], \quad (65)$$

where $s_e(\boldsymbol{u}, \boldsymbol{v}) = \| e(\boldsymbol{u}) - e(\boldsymbol{v}) \|_2^2$ is the squared Euclidean distance for any $\boldsymbol{u}, \boldsymbol{v} \in \mathcal{X}$.

Introducing the supervised contrastive loss to continual learning is novel, as existing methods often harness its ability to create a compact representation space, thereby mitigating representation overlapping in class incremental learning [65, 30, 9, 96]. However, in this work, the primary motivation for using the supervised contrastive loss to facilitate domain alignment lies in its optimal solution [29, 110, 19]. This optimal point referred to as *neural collapse* [47, 69, 111, 103], where the embeddings of the same class collapse to the same point, and those of different classes are sparsely distributed. It is easy to envision that when *the optimal state of the supervised constrastive loss* is attained across different domains, it concurrently achieves *perfect domain alignment*.

The loss function that fosters stability and domain alignment in the encoder $e$ can be summarized as follows:

$$\mathcal{L}_{\text{enc}}(e) = -\lambda_d V_d(d, e, \overset{\circ}{\Omega}) + \lambda_p V_p(e) + \lambda_s V_s(e). \tag{66}$$

Here, two hyper-parameters, $\lambda_p$ and $\lambda_s$, balance the influence of each individual loss on the encoder's embedding distribution. In practice, the final performance of UDIL is not significantly affected by the values of $\lambda_p$ and $\lambda_s$, and a wide range of values for these parameters can yield reasonable results.

## C.4  Evaluation Metrics

In continual learning, many evaluation metrics are based on the **Accuracy Matrix $\boldsymbol{R} \in \mathbb{R}^{T \times T}$**, where $T$ represents the total number of tasks (domains). In the accuracy matrix $\boldsymbol{R}$, the entry $R_{i,j}$ corresponds to the accuracy of the model when evaluated on task $j$ *after* training on task $i$. With this definition in mind, we primarily focus on the following specific metrics:

**Average Accuracy (Avg. Acc.)** up until domain $t$ represents the average accuracy of the first $t$ domains after training on these domains. We denote it as $A_t$ and define it as follows:

$$A_t \triangleq \tfrac{1}{t} \sum_{i=1}^{t} R_{t,i}. \tag{67}$$

In most of the continual learning literature, the final average accuracy $A_T$ is usually reported. In our paper, this metric is reported in the column labeled "**overall**". The average accuracy of a model is a crucial metric as it directly corresponds to the primary optimization goal of minimizing the error on all domains, as defined in Eqn. 3.

Additionally, to better illustrate the learning (and forgetting) process of a model across multiple domains, we propose the use of the "**Avg. of Avg. Acc.**" metric $A_{t_1:t_2}$, which represents the average of average accuracies for a consecutive range of domains starting from domain $t_1$ and ending at domain $t_2$. Specifically, we define this metric as follows:

$$A_{t_1:t_2} \triangleq \tfrac{1}{t_2 - t_1 + 1} \sum_{i=t_1}^{t_2} A_i. \tag{68}$$

This metric provides a condensed representation of the trend in accuracy variation compared to directly displaying the entire series of average accuracies $\{A_1, A_2, \cdots, A_T\}$. We report this Avg. of Avg. Acc. metric in all tables (except in Table 2 due to the limit of space).

**Average Forgetting (i.e., 'Forgetting' in the main paper)** defines the average of the largest drop of accuracy for each domain up till domain $t$. We denote this metric as $F_t$ and define it as follows:

$$F_t \triangleq \tfrac{1}{t-1} \sum_{j=1}^{t-1} f_t(j), \tag{69}$$

where $f_t(j)$ is the forgetting on domain $j$ after the model completes the training on domain $t$, which is defined as:

$$f_t(j) \triangleq \max_{l \in [t-1]} \{R_{l,j} - R_{t,j}\}. \tag{70}$$

Typically, the average forgetting is reported after training on the last domain $T$. Measuring forgetting is of great practical significance, especially when two models have similar average accuracies. It indicates how a model balances *stability* and *plasticity*. If a model $\mathcal{P}$ achieves a reasonable final average accuracy across different domains but exhibits high forgetting, we can conclude that this model has high plasticity and low stability. It quickly adapts to new domains but at the expense of performance on past domains. On the other hand, if another model $\mathcal{S}$ has a similar average accuracy to $\mathcal{P}$ but significantly lower average forgetting, we can infer that the model $\mathcal{S}$ has high stability and low plasticity. It sacrifices performance on recent domains to maintain a reasonable performance on past domains. Hence, to gain a comprehensive understanding of model performance, we focus on evaluating two key metrics: *Avg. Acc.* and *Forgetting*. These metrics provide insights into how models balance stability and plasticity and allow us to assess their overall performance across different domains.

**Forward Transfer** $W_t$ quantifies the extent to which learning from past $t-1$ domains contributes to the performance on the next domain $t$. It is defined as follows:

$$W_t \triangleq \tfrac{1}{t-1} \sum_{i=2}^{t} R_{i-1,i} - r_i, \tag{71}$$

where $r_i$ is the accuracy of a randomly initialized model evaluated on domain $i$. For domain incremental learning, where the model does not have access to future domain data and does not explicitly optimize for higher Forward Transfer, the results of this metric are *typically random. Therefore, we do not report this metric in the complete tables presented in this section.*

### C.5 Introduction to Baselines

We compare UDIL with the state-of-the-art continual learning methods that are either specifically designed for domain incremental learning or can be easily adapted to the domain incremental learning setting. Exemplar-free baselines include online Elastic Weight Consolidation (**oEWC**) [81], Synaptic Intelligence (**SI**) [105], and Learning without Forgetting (**LwF**) [52]. Memory-based domain incremental learning baselines include Gradient Episodic Memory (**GEM**) [58], Averaged Gradient Episodic Memory (**A-GEM**) [10], Experience Replay (**ER**) [75], Dark Experience Replay (**DER++**) [8], and two recent methods, Complementary Learning System based Experience Replay (**CLS-ER**) [4] and Error Sensitivity Modulation based Experience Replay (**ESM-ER**) [80]. In addition, we implement the fine-tuning (**Fine-tune**) [52] and joint-training (**Joint**) as the performance lower bound and upper bound (Oracle). Here we provide a short description of the primary idea of the memory-based domain incremental learning baselines.

- **GEM** [58]: The baseline method that uses the memory to provide additional optimization constraints during learning the current domain. Specifically, the update of the model cannot point towards the direction at which the loss of any exemplar increases.
- **A-GEM** [10]: The improved baseline method where the constraints of GEM are averaged as one, which shortens the computational time significantly.
- **ER** [75]: The fundamental memory-based domain incremental learning framework where the mini-batch of the memory is regularly replayed with the current domain data.
- **DER++** [8]: A simple yet effective replay-based method where an additional logits distillation (dubbed "dark experience replay") is applied compared to the vanilla ER.
- **CLS-ER** [4]: A complementary learning system inspired replay method, where two exponential moving average models are used to serve as the semantic memory, which provides the logits distillation target during training.
- **ESM-ER** [80]: An improved version of CLS-ER, where the effect of large errors when learning the current domain is reduced, dubbed "error sensitivity modulation".

### C.6 Training Schemes

**Training Process.** For each group of experiments, we run three rounds with different seeds and report the mean and standard deviation of the results. We follow the optimal configurations (epochs and learning rate) stated in [8, 80] for the baselines in *P-MNIST* and *R-MNIST* dataset. For *HD-Balls* and *Seq-CORe50*, we first search for the optimal training configuration for the joint learning, and then grid-search the configuration in a small range near it for the baselines listed above. For our UDIL framework, as it involves adversarial training for the domain embedding alignment, we typically need a configuration that has larger number of epochs and smaller learning rate. We use a simple grid search to achieve the optimal configuration for it as well.

**Model Architectures.** For the baseline methods and UDIL in the same dataset, we adopt the same backbone neural architectures to ensure fair comparison. In *HD-Balls*, we adopt the same multi-layer perceptron with the same separation of encoder and decoder as in CIDA [94], where the hidden dimension is set to 800. In *P-MNIST* and *R-MNIST*, we adopt the same multi-layer perceptron architecture as in DER++ [8] with hidden dimension set to 800 as well. In *Seq-CORe50*, we use the ResNet18 [33] as our backbone architecture for all the methods, where the layers before the final average pooling are treated as the encoder $e$, and the remaining part is treated as the predictor $p$.

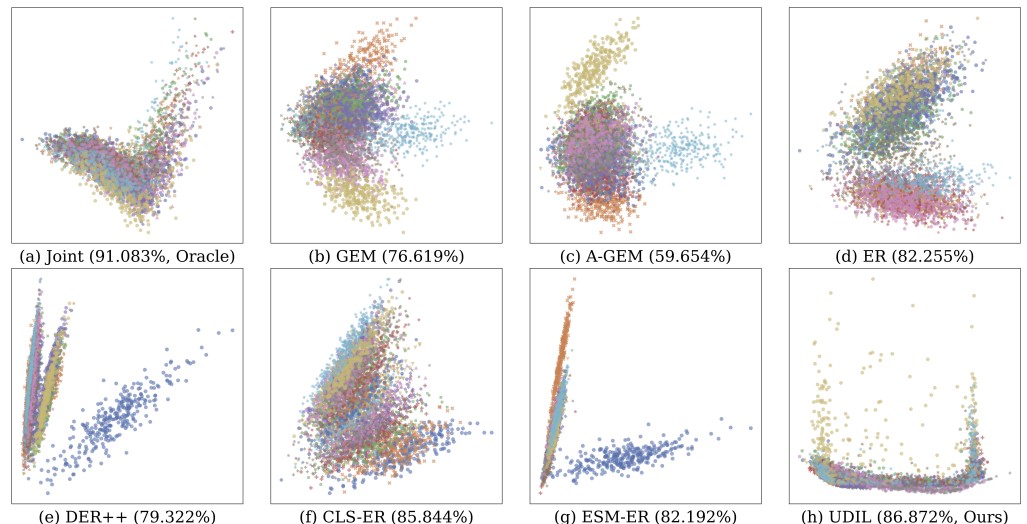

| (a) Joint (91.083%, Oracle) | (b) GEM (76.619%) | (c) A-GEM (59.654%) | (d) ER (82.255%) |
| (e) DER++ (79.322%) | (f) CLS-ER (85.844%) | (g) ESM-ER (82.192%) | (h) UDIL (86.872%, Ours) |

Figure 3: More Results on *HD-Balls*. Data is colored according to domain ID. All data is plotted after PCA [6]. **(a-h)** Accuracy and embeddings learned by Joint (oracle), UDIL, and six baselines with memory. Joint, as the *oracle*, naturally aligns different domains, and UDIL outperforms all baselines in terms of embedding alignment and accuracy.

**Hyperparameter Setting.** For setting the hyper-parameter embedding alignment strength coefficient $\lambda_d$ and parameter $C$ that models the combined effect of VC-dimension $d$ and error tolerance $\delta$, we use grid search for each dataset, where the range $\lambda_d \in [0.01, 100]$ and $C \in [0, 1000]$ are used.

# D Additional Empirical Results

This section presents additional empirical results of the UDIL algorithm. Sec. D.1 will show the additional results on different constraints with varying memory sizes. Sec. D.2 provides additional qualitative results: visualization of embedding distributions, to showcase the importance of the embedding alignment across domains.

## D.1 Empirical Results on Varying Memory Sizes

Here we present additional empirical results to validate the effectiveness of our UDIL framework using varying memory sizes. The evaluation is conducted on three real-world datasets, as shown in Table 5, Table 6, and Table 7. By increasing the memory size from $400$ to $800$ in Table 5 and 6 and from $500$ to $1000$ in Table 7, we can investigate the impact of having access to a larger pool of past experiences on the continual learning performance, which might occur when the constraint on memory capacity is relaxed. This allows us to study the benefits of a more extensive memory in terms of knowledge retention and performance improvement. On the other hand, by further decreasing the memory size to the extreme of $200$ in Table 5 and 6, we can explore the consequences of severely limited memory capacity. This scenario simulates situations where memory constraints are extremely tight, and the model can only retain a small fraction of past domain data, for example, a model deployed on edge devices. To ensure a fair comparison, here we use the same best configuration found in the main body of this work.

The results in all three tables demonstrate a clear advantage of our UDIL framework when the memory size is limited. In *P-MNIST* and *R-MNIST*, when the memory size $|\mathcal{M}| = 200$, the overall performance of UDIL reaches $91.483\%$ and $82.796\%$ respectively, which outperforms the second best model DER++ by $0.757\%$ and a remarkable $6.125\%$. In *Seq-CORe50*, when the memory size $|\mathcal{M}| = 500$ is set, UDIL holds a $3.474\%$ lead compared to the second best result. When the memory size is larger, the gap between UDIL and the baseline models is smaller. This is because when the memory constraint is relaxed, all the continual learning models should be at least closer to the performance upper bound, i.e., joint learning or 'Joint (Oracle)' in the tables, causing the

indistinguishable results among each other. Apparently, DER++ favors larger memory more than UDIL, while UDIL can still maintain a narrow lead in the large scale dataset *Seq-CORe50*.

## D.2 Visualization of Embedding Spaces

Here we provide more embedding space visualization results for the baselines with the utilization of memory, shown in Fig. 3. As one of the primary objectives of our algorithm, embedding space alignment across multiple domains naturally follows the pattern shown in the joint learning and therefore leads to a higher performance.

Table 5: **Performances (%) evaluated on *P-MNIST*.** Average Accuracy (Avg. Acc.) and Forgetting are reported to measure the methods' performance. "↑" and "↓" mean higher and lower numbers are better, respectively. We use **boldface** and underlining to denote the best and the second-best performance, respectively. We use "-" to denote "not appliable".

| Method | Buffer | $\mathcal{D}_{1:5}$ | $\mathcal{D}_{6:10}$ | $\mathcal{D}_{11:15}$ | $\mathcal{D}_{16:20}$ | Overall | |
| --- | --- | --- | --- | --- | --- | --- | --- |
| | | Avg. Acc (↑) | | | | Avg. Acc (↑) | Forgetting (↓) |
| Fine-tune | - | $92.506_{\pm2.062}$ | $87.088_{\pm1.337}$ | $81.295_{\pm2.372}$ | $72.807_{\pm1.817}$ | $70.102_{\pm2.945}$ | $27.522_{\pm3.042}$ |
| oEWC [81] | - | $92.415_{\pm0.816}$ | $87.988_{\pm1.607}$ | $83.098_{\pm1.843}$ | $78.670_{\pm0.902}$ | $78.476_{\pm1.223}$ | $18.068_{\pm1.321}$ |
| SI [105] | - | $92.282_{\pm0.862}$ | $87.986_{\pm1.622}$ | $83.698_{\pm1.220}$ | $79.669_{\pm0.709}$ | $79.045_{\pm1.357}$ | $17.409_{\pm1.446}$ |
| LwF [52] | - | $95.025_{\pm0.487}$ | $91.402_{\pm1.546}$ | $83.984_{\pm2.103}$ | $76.046_{\pm2.004}$ | $73.545_{\pm2.646}$ | $24.556_{\pm2.789}$ |
| GEM [58] | | $93.310_{\pm0.374}$ | $91.900_{\pm0.456}$ | $89.813_{\pm0.914}$ | $87.251_{\pm0.524}$ | $86.729_{\pm0.203}$ | $9.430_{\pm0.156}$ |
| A-GEM [10] | | $93.326_{\pm0.363}$ | $91.466_{\pm0.605}$ | $89.048_{\pm1.005}$ | $86.518_{\pm0.604}$ | $85.712_{\pm0.228}$ | $10.485_{\pm0.196}$ |
| ER [75] | | $94.087_{\pm0.762}$ | $92.397_{\pm0.464}$ | $89.999_{\pm1.060}$ | $87.492_{\pm0.448}$ | $86.963_{\pm0.303}$ | $9.273_{\pm0.255}$ |
| DER++ [8] | 200 | $94.708_{\pm0.451}$ | $\underline{94.582}_{\pm0.158}$ | $\underline{93.271}_{\pm0.585}$ | $90.980_{\pm0.610}$ | $90.333_{\pm0.587}$ | $6.110_{\pm0.545}$ |
| CLS-ER [4] | | $94.761_{\pm0.340}$ | $93.943_{\pm0.197}$ | $92.725_{\pm0.566}$ | $91.150_{\pm0.357}$ | $\underline{90.726}_{\pm0.218}$ | $\underline{5.428}_{\pm0.252}$ |
| ESM-ER [80] | | $\underline{95.198}_{\pm0.236}$ | $94.029_{\pm0.427}$ | $91.710_{\pm1.056}$ | $88.181_{\pm1.021}$ | $86.851_{\pm0.858}$ | $10.007_{\pm0.864}$ |
| UDIL (Ours) | | $\mathbf{95.747}_{\pm0.486}$ | $\mathbf{94.695}_{\pm0.256}$ | $\mathbf{93.756}_{\pm0.343}$ | $\mathbf{92.254}_{\pm0.564}$ | $\mathbf{91.483}_{\pm0.270}$ | $\mathbf{4.399}_{\pm0.314}$ |
| GEM [58] | | $93.557_{\pm0.225}$ | $92.635_{\pm0.306}$ | $91.246_{\pm0.492}$ | $89.565_{\pm0.342}$ | $89.097_{\pm0.149}$ | $6.975_{\pm0.167}$ |
| A-GEM [10] | | $93.432_{\pm0.333}$ | $92.064_{\pm0.439}$ | $90.038_{\pm0.726}$ | $87.988_{\pm0.335}$ | $87.560_{\pm0.087}$ | $8.577_{\pm0.053}$ |
| ER [75] | | $93.525_{\pm1.101}$ | $91.649_{\pm0.362}$ | $90.426_{\pm0.456}$ | $88.728_{\pm0.353}$ | $88.339_{\pm0.044}$ | $7.180_{\pm0.029}$ |
| DER++ [8] | 400 | $94.952_{\pm0.403}$ | $\mathbf{95.089}_{\pm0.075}$ | $\mathbf{94.458}_{\pm0.328}$ | $\mathbf{93.257}_{\pm0.249}$ | $\mathbf{92.950}_{\pm0.361}$ | $\underline{3.378}_{\pm0.245}$ |
| CLS-ER [4] | | $94.262_{\pm0.649}$ | $93.195_{\pm0.148}$ | $92.623_{\pm0.195}$ | $91.839_{\pm0.187}$ | $91.598_{\pm0.117}$ | $3.795_{\pm0.144}$ |
| ESM-ER [80] | | $\underline{95.413}_{\pm0.139}$ | $94.654_{\pm0.314}$ | $93.353_{\pm0.588}$ | $91.022_{\pm0.781}$ | $89.829_{\pm0.698}$ | $6.888_{\pm0.738}$ |
| UDIL (Ours) | | $\mathbf{95.992}_{\pm0.349}$ | $\underline{95.026}_{\pm0.250}$ | $\underline{94.212}_{\pm0.280}$ | $\underline{93.094}_{\pm0.326}$ | $\underline{92.666}_{\pm0.108}$ | $\mathbf{2.853}_{\pm0.107}$ |
| GEM [58] | | $93.717_{\pm0.177}$ | $93.116_{\pm0.206}$ | $92.166_{\pm0.335}$ | $91.076_{\pm0.342}$ | $90.609_{\pm0.364}$ | $5.393_{\pm0.417}$ |
| A-GEM [10] | | $93.612_{\pm0.241}$ | $92.523_{\pm0.375}$ | $90.718_{\pm0.739}$ | $88.543_{\pm0.391}$ | $88.020_{\pm0.851}$ | $8.081_{\pm0.867}$ |
| ER [75] | | $93.827_{\pm0.871}$ | $92.457_{\pm0.217}$ | $91.688_{\pm0.277}$ | $90.617_{\pm0.289}$ | $90.252_{\pm0.056}$ | $5.188_{\pm0.045}$ |
| DER++ [8] | 800 | $95.295_{\pm0.317}$ | $\mathbf{95.539}_{\pm0.041}$ | $\mathbf{95.099}_{\pm0.187}$ | $\mathbf{94.423}_{\pm0.151}$ | $\mathbf{94.227}_{\pm0.261}$ | $\underline{2.106}_{\pm0.161}$ |
| CLS-ER [4] | | $94.463_{\pm0.537}$ | $93.567_{\pm0.093}$ | $93.182_{\pm0.137}$ | $92.744_{\pm0.112}$ | $92.578_{\pm0.152}$ | $2.803_{\pm0.183}$ |
| ESM-ER [80] | | $\underline{95.567}_{\pm0.150}$ | $95.136_{\pm0.202}$ | $94.301_{\pm0.347}$ | $92.981_{\pm0.397}$ | $92.408_{\pm0.387}$ | $4.170_{\pm0.357}$ |
| UDIL (Ours) | | $\mathbf{96.082}_{\pm0.313}$ | $\underline{95.207}_{\pm0.196}$ | $\underline{94.642}_{\pm0.156}$ | $\underline{93.997}_{\pm0.194}$ | $\underline{93.724}_{\pm0.043}$ | $\mathbf{1.633}_{\pm0.035}$ |
| Joint (Oracle) | $\infty$ | - | - | - | - | $96.368_{\pm0.042}$ | - |

Table 6: **Performances (%) evaluated on *R-MNIST*.** Average Accuracy (Avg. Acc.) and Forgetting are reported to measure the methods' performance. "↑" and "↓" mean higher and lower numbers are better, respectively. We use **boldface** and underlining to denote the best and the second-best performance, respectively. We use "-" to denote "not appliable".

| Method | Buffer | $\mathcal{D}_{1:5}$ | $\mathcal{D}_{6:10}$ | $\mathcal{D}_{11:15}$ | $\mathcal{D}_{16:20}$ | Overall | |
| | | Avg. Acc (↑) | | | | Avg. Acc (↑) | Forgetting (↓) |
|---|---|---|---|---|---|---|---|
| Fine-tune | - | $92.961_{\pm2.683}$ | $76.617_{\pm8.011}$ | $60.212_{\pm3.688}$ | $49.793_{\pm1.552}$ | $47.803_{\pm1.703}$ | $52.281_{\pm1.797}$ |
| oEWC [81] | - | $91.765_{\pm2.286}$ | $76.226_{\pm7.622}$ | $60.320_{\pm3.892}$ | $50.505_{\pm1.772}$ | $48.203_{\pm0.827}$ | $51.181_{\pm0.867}$ |
| SI [105] | - | $91.867_{\pm2.272}$ | $76.801_{\pm7.391}$ | $60.956_{\pm3.504}$ | $50.301_{\pm1.538}$ | $48.251_{\pm1.381}$ | $51.053_{\pm1.507}$ |
| LwF [52] | - | $95.174_{\pm1.154}$ | $83.044_{\pm5.935}$ | $65.899_{\pm4.061}$ | $55.980_{\pm1.296}$ | $54.709_{\pm0.515}$ | $45.473_{\pm0.565}$ |
| GEM [58] | | $93.441_{\pm0.610}$ | $88.620_{\pm2.381}$ | $81.034_{\pm2.704}$ | $73.112_{\pm1.922}$ | $70.545_{\pm0.623}$ | $27.684_{\pm0.645}$ |
| A-GEM [10] | | $92.667_{\pm1.352}$ | $82.772_{\pm5.503}$ | $70.579_{\pm4.028}$ | $60.462_{\pm2.001}$ | $57.958_{\pm0.579}$ | $40.969_{\pm0.580}$ |
| ER [75] | | $94.705_{\pm0.790}$ | $89.171_{\pm2.883}$ | $79.962_{\pm3.365}$ | $71.787_{\pm1.608}$ | $69.627_{\pm0.911}$ | $28.749_{\pm0.993}$ |
| DER++ [8] | 200 | $94.904_{\pm0.414}$ | $91.637_{\pm1.871}$ | $84.915_{\pm2.315}$ | $78.373_{\pm1.244}$ | $76.671_{\pm0.391}$ | $21.743_{\pm0.409}$ |
| CLS-ER [4] | | $95.131_{\pm0.523}$ | $91.421_{\pm1.732}$ | $84.773_{\pm2.665}$ | $77.733_{\pm1.480}$ | $75.609_{\pm0.418}$ | $22.483_{\pm0.456}$ |
| ESM-ER [80] | | $\mathbf{95.378_{\pm0.531}}$ | $90.800_{\pm2.528}$ | $83.438_{\pm2.581}$ | $76.987_{\pm1.219}$ | $75.203_{\pm0.143}$ | $23.564_{\pm0.157}$ |
| UDIL (Ours) | | $\underline{95.097_{\pm0.447}}$ | $\mathbf{93.101_{\pm1.305}}$ | $\mathbf{89.194_{\pm1.472}}$ | $\mathbf{84.704_{\pm1.722}}$ | $\mathbf{82.796_{\pm1.882}}$ | $\mathbf{12.971_{\pm2.389}}$ |
| GEM [58] | | $93.842_{\pm0.313}$ | $90.663_{\pm1.856}$ | $85.392_{\pm1.856}$ | $79.061_{\pm1.578}$ | $76.619_{\pm0.581}$ | $21.289_{\pm0.579}$ |
| A-GEM [10] | | $92.820_{\pm1.274}$ | $83.564_{\pm5.024}$ | $72.616_{\pm3.865}$ | $62.223_{\pm2.081}$ | $59.654_{\pm0.122}$ | $39.196_{\pm0.171}$ |
| ER [75] | | $94.916_{\pm0.457}$ | $91.491_{\pm1.878}$ | $86.029_{\pm2.176}$ | $78.688_{\pm1.323}$ | $76.794_{\pm0.696}$ | $20.696_{\pm0.744}$ |
| DER++ [8] | 400 | $95.246_{\pm0.228}$ | $93.627_{\pm1.147}$ | $90.011_{\pm1.289}$ | $85.601_{\pm0.982}$ | $84.258_{\pm0.544}$ | $13.692_{\pm0.560}$ |
| CLS-ER [4] | | $95.233_{\pm0.271}$ | $92.740_{\pm1.268}$ | $89.111_{\pm1.305}$ | $83.678_{\pm1.388}$ | $81.771_{\pm0.354}$ | $15.455_{\pm0.356}$ |
| ESM-ER [80] | | $\mathbf{95.825_{\pm0.303}}$ | $93.378_{\pm1.480}$ | $89.290_{\pm1.604}$ | $83.868_{\pm1.163}$ | $82.192_{\pm0.164}$ | $16.195_{\pm0.150}$ |
| UDIL (Ours) | | $\underline{95.274_{\pm0.469}}$ | $\mathbf{94.043_{\pm0.759}}$ | $\mathbf{91.511_{\pm0.990}}$ | $\mathbf{87.809_{\pm0.849}}$ | $\mathbf{86.635_{\pm0.686}}$ | $\mathbf{8.506_{\pm1.181}}$ |
| GEM [58] | | $94.212_{\pm0.322}$ | $92.482_{\pm1.125}$ | $89.191_{\pm1.346}$ | $84.866_{\pm1.317}$ | $82.772_{\pm1.079}$ | $14.781_{\pm1.104}$ |
| A-GEM [10] | | $92.902_{\pm1.194}$ | $84.611_{\pm4.451}$ | $75.150_{\pm3.421}$ | $64.510_{\pm2.437}$ | $61.240_{\pm1.026}$ | $37.528_{\pm1.089}$ |
| ER [75] | | $95.144_{\pm0.281}$ | $92.997_{\pm1.195}$ | $89.319_{\pm1.365}$ | $84.352_{\pm1.681}$ | $81.877_{\pm1.157}$ | $15.285_{\pm1.196}$ |
| DER++ [8] | 800 | $\underline{95.496_{\pm0.261}}$ | $\mathbf{94.960_{\pm0.568}}$ | $\mathbf{93.013_{\pm0.689}}$ | $\mathbf{90.820_{\pm0.687}}$ | $\mathbf{89.746_{\pm0.356}}$ | $\underline{7.821_{\pm0.371}}$ |
| CLS-ER [4] | | $95.462_{\pm0.174}$ | $93.927_{\pm0.881}$ | $91.275_{\pm0.930}$ | $87.816_{\pm0.988}$ | $86.418_{\pm0.215}$ | $10.598_{\pm0.228}$ |
| ESM-ER [80] | | $\mathbf{96.086_{\pm0.361}}$ | $94.746_{\pm0.915}$ | $92.393_{\pm0.974}$ | $89.745_{\pm0.712}$ | $88.662_{\pm0.263}$ | $9.409_{\pm0.255}$ |
| UDIL (Ours) | | $95.354_{\pm0.480}$ | $\underline{94.711_{\pm0.563}}$ | $\underline{92.776_{\pm0.695}}$ | $\underline{90.399_{\pm0.755}}$ | $\underline{89.191_{\pm0.685}}$ | $\mathbf{6.351_{\pm1.304}}$ |
| Joint (Oracle) | ∞ | - | - | - | - | $97.150_{\pm0.036}$ | - |

Table 7: **Performances (%) evaluated on *Seq-CORe50*.** Avg. Acc. and Forgetting are reported to measure the methods' performance. "↑" and "↓" mean higher and lower numbers are better, respectively. We use **boldface** and underlining to denote the best and the second-best performance, respectively. We use "-" to denote "not appliable" and "⋆" to denote out-of-memory (*OOM*) error when running the experiments.

| Method | Buffer | $\mathcal{D}_{1:3}$ | $\mathcal{D}_{4:6}$ | $\mathcal{D}_{7:9}$ | $\mathcal{D}_{10:11}$ | Overall | |
| --- | --- | --- | --- | --- | --- | --- | --- |
| | | Avg. Acc (↑) | | | | Avg. Acc (↑) | Forgetting (↓) |
| Fine-tune | - | $73.707_{\pm13.144}$ | $34.551_{\pm1.254}$ | $29.406_{\pm2.579}$ | $28.689_{\pm3.144}$ | $31.832_{\pm1.034}$ | $73.296_{\pm1.399}$ |
| oEWC [81] | - | $74.567_{\pm13.360}$ | $35.915_{\pm0.260}$ | $30.174_{\pm3.195}$ | $28.291_{\pm2.522}$ | $30.813_{\pm1.154}$ | $74.563_{\pm0.937}$ |
| SI [105] | - | $74.661_{\pm14.162}$ | $34.345_{\pm1.001}$ | $30.127_{\pm2.971}$ | $28.839_{\pm3.631}$ | $32.469_{\pm1.315}$ | $73.144_{\pm1.588}$ |
| LwF [52] | - | $80.383_{\pm10.190}$ | $28.357_{\pm1.143}$ | $31.386_{\pm0.787}$ | $28.711_{\pm2.981}$ | $31.692_{\pm0.768}$ | $72.990_{\pm1.350}$ |
| GEM [58] | | $79.852_{\pm6.864}$ | $38.961_{\pm1.718}$ | $39.258_{\pm2.614}$ | $36.859_{\pm0.842}$ | $37.701_{\pm0.273}$ | $22.724_{\pm1.554}$ |
| A-GEM [10] | | $80.348_{\pm9.394}$ | $41.472_{\pm3.394}$ | $43.213_{\pm1.542}$ | $39.181_{\pm3.999}$ | $43.181_{\pm2.025}$ | $33.775_{\pm3.003}$ |
| ER [75] | | $90.838_{\pm2.177}$ | $79.343_{\pm2.699}$ | $68.151_{\pm0.226}$ | $65.034_{\pm1.571}$ | $66.605_{\pm0.214}$ | $32.750_{\pm0.455}$ |
| DER++ [8] | 500 | $\underline{92.444}_{\pm1.764}$ | $\underline{88.652}_{\pm1.854}$ | $\underline{80.391}_{\pm0.107}$ | $\underline{78.038}_{\pm0.591}$ | $\underline{78.629}_{\pm0.753}$ | $\underline{21.910}_{\pm1.094}$ |
| CLS-ER [4] | | $89.834_{\pm1.323}$ | $78.909_{\pm1.724}$ | $70.591_{\pm0.322}$ | $\star$ | $\star$ | $\star$ |
| ESM-ER [80] | | $84.905_{\pm6.471}$ | $51.905_{\pm3.257}$ | $53.815_{\pm1.770}$ | $50.178_{\pm2.574}$ | $52.751_{\pm1.296}$ | $25.444_{\pm0.580}$ |
| UDIL (Ours) | | $\mathbf{98.152}_{\pm1.665}$ | $\mathbf{89.814}_{\pm2.302}$ | $\mathbf{83.052}_{\pm0.151}$ | $\mathbf{81.547}_{\pm0.269}$ | $\mathbf{82.103}_{\pm0.279}$ | $\mathbf{19.589}_{\pm0.303}$ |
| GEM [58] | | $78.717_{\pm4.831}$ | $43.269_{\pm3.419}$ | $40.908_{\pm2.200}$ | $40.408_{\pm1.168}$ | $41.576_{\pm1.599}$ | $18.537_{\pm1.237}$ |
| A-GEM [10] | | $78.917_{\pm8.984}$ | $41.172_{\pm4.293}$ | $44.576_{\pm1.701}$ | $38.960_{\pm3.867}$ | $42.827_{\pm1.659}$ | $33.800_{\pm1.847}$ |
| ER [75] | | $\underline{90.048}_{\pm2.699}$ | $84.668_{\pm1.988}$ | $77.561_{\pm1.281}$ | $72.268_{\pm0.720}$ | $72.988_{\pm0.566}$ | $25.997_{\pm0.694}$ |
| DER++ [8] | 1000 | $89.510_{\pm5.726}$ | $\underline{92.492}_{\pm0.902}$ | $\underline{88.883}_{\pm0.794}$ | $\underline{86.108}_{\pm0.284}$ | $\underline{86.392}_{\pm0.714}$ | $\underline{13.128}_{\pm0.474}$ |
| CLS-ER [4] | | $92.004_{\pm0.894}$ | $85.044_{\pm1.276}$ | $\star$ | $\star$ | $\star$ | $\star$ |
| ESM-ER [80] | | $85.120_{\pm4.339}$ | $54.852_{\pm5.511}$ | $61.714_{\pm1.840}$ | $55.098_{\pm3.834}$ | $58.932_{\pm0.959}$ | $20.134_{\pm0.643}$ |
| UDIL (Ours) | | $\mathbf{98.648}_{\pm1.174}$ | $\mathbf{93.447}_{\pm1.111}$ | $\mathbf{90.545}_{\pm0.705}$ | $\mathbf{87.923}_{\pm0.232}$ | $\mathbf{88.155}_{\pm0.445}$ | $\mathbf{12.882}_{\pm0.460}$ |
| Joint (Oracle) | $\infty$ | - | - | - | - | $99.137_{\pm0.049}$ | - |

