{95.747}_{\pm\mathbf{0.486}}$ | $\mathbf{94.695}_{\pm\mathbf{0.256}}$ | $\mathbf{93.756}_{\pm\mathbf{0.343}}$ | $\mathbf{92.254}_{\pm\mathbf{0.564}}$ | $\mathbf{91.483}_{\pm\mathbf{0.270}}$ | $\mathbf{4.399}_{\pm\mathbf{0.314}}$ |
| GEM [34] | | $93.557_{\pm0.225}$ | $92.635_{\pm0.306}$ | $91.246_{\pm0.492}$ | $89.565_{\pm0.342}$ | $89.097_{\pm0.149}$ | $6.975_{\pm0.167}$ |
| A-GEM [8] | | $93.432_{\pm0.333}$ | $92.064_{\pm0.439}$ | $90.038_{\pm0.726}$ | $87.988_{\pm0.335}$ | $87.560_{\pm0.087}$ | $8.577_{\pm0.053}$ |
| ER [46] | | $93.525_{\pm1.101}$ | $91.649_{\pm0.362}$ | $90.426_{\pm0.456}$ | $88.728_{\pm0.353}$ | $88.339_{\pm0.044}$ | $7.180_{\pm0.029}$ |
| DER++ [6] | 400 | $94.952_{\pm0.403}$ | $\mathbf{95.089}_{\pm\mathbf{0.075}}$ | $\mathbf{94.458}_{\pm\mathbf{0.328}}$ | $\mathbf{93.257}_{\pm\mathbf{0.249}}$ | $\mathbf{92.950}_{\pm\mathbf{0.361}}$ | $\underline{3.378}_{\pm0.245}$ |
| CLS-ER [3] | | $94.262_{\pm0.649}$ | $93.195_{\pm0.148}$ | $92.623_{\pm0.195}$ | $91.839_{\pm0.187}$ | $91.598_{\pm0.117}$ | $3.795_{\pm0.144}$ |
| ESM-ER [50] | | $\underline{95.413}_{\pm0.139}$ | $\underline{94.654}_{\pm0.314}$ | $93.353_{\pm0.588}$ | $91.022_{\pm0.781}$ | $89.829_{\pm0.698}$ | $6.888_{\pm0.738}$ |
| UDIL (Ours) | | $\mathbf{95.992}_{\pm\mathbf{0.349}}$ | $95.026_{\pm0.250}$ | $\underline{94.212}_{\pm0.280}$ | $\underline{93.094}_{\pm0.326}$ | $\underline{92.666}_{\pm0.108}$ | $\mathbf{2.853}_{\pm\mathbf{0.107}}$ |
| GEM [34] | | $93.717_{\pm0.177}$ | $93.116_{\pm0.206}$ | $92.166_{\pm0.335}$ | $91.076_{\pm0.342}$ | $90.609_{\pm0.364}$ | $5.393_{\pm0.417}$ |
| A-GEM [8] | | $93.612_{\pm0.241}$ | $92.523_{\pm0.375}$ | $90.718_{\pm0.739}$ | $88.543_{\pm0.391}$ | $88.020_{\pm0.851}$ | $8.081_{\pm0.867}$ |
| ER [46] | | $93.827_{\pm0.871}$ | $92.457_{\pm0.217}$ | $91.688_{\pm0.277}$ | $90.617_{\pm0.289}$ | $90.252_{\pm0.056}$ | $5.188_{\pm0.045}$ |
| DER++ [6] | 800 | $95.295_{\pm0.317}$ | $\mathbf{95.539}_{\pm\mathbf{0.041}}$ | $\mathbf{95.099}_{\pm\mathbf{0.187}}$ | $\mathbf{94.423}_{\pm\mathbf{0.151}}$ | $\mathbf{94.227}_{\pm\mathbf{0.261}}$ | $\underline{2.106}_{\pm0.161}$ |
| CLS-ER [3] | | $94.463_{\pm0.537}$ | $93.567_{\pm0.093}$ | $93.182_{\pm0.137}$ | $92.744_{\pm0.112}$ | $92.578_{\pm0.152}$ | $2.803_{\pm0.183}$ |
| ESM-ER [50] | | $\underline{95.567}_{\pm0.150}$ | $95.136_{\pm0.202}$ | $94.301_{\pm0.347}$ | $92.981_{\pm0.397}$ | $92.408_{\pm0.387}$ | $4.170_{\pm0.357}$ |
| UDIL (Ours) | | $\mathbf{96.082}_{\pm\mathbf{0.313}}$ | $\underline{95.207}_{\pm0.196}$ | $\underline{94.642}_{\pm0.156}$ | $\underline{93.997}_{\pm0.194}$ | $\underline{93.724}_{\pm0.043}$ | $\mathbf{1.633}_{\pm\mathbf{0.035}}$ |
| Joint (Oracle) | $\infty$ | - | - | - | - | $96.368_{\pm0.042}$ | - |