# OpenReview forum: "A Unified Approach to Domain Incremental Learning with Memory: Theory and Algorithm"
_NeurIPS.cc/2023/Conference — NeurIPS 2023 poster_

### Official Review · Reviewer_4VUe · 2023-07-05

**Soundness:** 4 excellent
**Presentation:** 4 excellent
**Contribution:** 4 excellent
**Rating:** 7
**Confidence:** 3

**Summary:**

This paper proposes a framework for domain incremental learning that unifies ideas from previous methods in this area. A theoretical analysis is first performed to formally state the problem of domain incremental learning. The theoretical formalization identifies three ways to bound the risk of error on a particular domain: 1) by a naive ERM-based bound, 2) by an intra-domain model-based bound, 3) by a cross-domain model-based bound. Based on the identified limitations of these three bound, the authors proposed a unified view of the generalization bound for domains observed in incremental learning. They propose to integrate these three bounds using coefficients that vary depending on the domains at hand. Existing methods in domain incremental learning are then categorized according to the components of the unified framework. In the second part of the paper, the authors propose a method where the coefficients are both learnable and dynamic. Experimental results on both toy data and benchmark datasets, along with comparison to SOTA methods in the area, are provided.

---Rebuttal---
I read the authors rebuttal. My review was already positive, so I left my score unchanged.

**Strengths:**

- Regarding originality, the paper presents originality both at the theoretical and experimental levels. Regarding theoretical originality, to the best of my knowledge the paper presents the first comprehensive analysis of generalization bounds in domain incremental learning while also categorizing SOTA methods in the area according to different views of these bounds. Regarding experimental originality, the paper presents a method that is dynamic and adaptable to relevant aspects such as the memory size and the domain relatedness in domain incremental learning.

- Regarding significance, the paper provides new insights on domain incremental learning through rigorous and extensive theoretical analysis. This is of high relevance for researchers in this subfield.

- Regarding quality and clarity, the paper is very well written and it is easy to follow. Extensive details are provided both in the paper and in the supplementary material.

**Weaknesses:**

- From the experimental results presented in Tables 2 and 3, the proposed method seems to be competitive but not significantly better than counterparts (especially DER++).

- Details on the computational complexity/cost of the method are not provided in the formulation of the method or in the experiments. Considering that the method needs to learn the coefficients, and therefore the problem in Eq. 13 is more difficult than the optimization problem in counterpart methods, what would be the extra-cost in terms of computation?

**Questions:**

Please refer to the "weaknesses" section for questions.

**Limitations:**

Limitations are properly addressed thorough the paper. There are not potential negative societal impacts of this work that need to be stated explicitly.

---

> ### Author Rebuttal · Authors · 2023-08-07
>
> Thank you for the constructive and encouraging comments as well as the insightful questions. We are glad that you find our work ``"original"`` at both ``"theoretical and experimental levels"``, our paper ``"very well-written and easy to follow"``, and acknowledge that our UDIL is ``"the first comprehensive analysis of generalization bounds in domain incremental learning"``. Below we address your questions one by one.
>
>
> **Q1. In Table 2 and 3, the proposed method seems to be competitive but not significantly better than counterparts (especially DER++).**
>
> This is an important question. For datasets Rotating-MNIST and Seq-CORe50, where different domains' data exhibits a more natural distributional shift, UDIL shines with notable improvements of 2 and 4 absolute accuracy points over the second best model, DER++. These improvements are significant, particularly considering the number of domains (20 in Rotating-MNIST) and the number of classes (50 per domain in Seq-CORe50) taken into account.
>
> For the toy dataset HD-Balls, our UDIL demonstrates superior performance, *outperforming DER++ by up to 7 absolute accuracy points* and surpassing the second best method, CLS-ER, by 1 absolute accuracy point. We suspect the smaller (but *still statistically significant*) margin over CLS-ER is due to the relatively simple data distribution of HD-Balls.
>
> For the three datasets above, we ran the corresponding hypothesis tests, and the $p$-values are in the range of $(1.23\times 10^{-3}, 5.24\times 10^{-3})$, much lower than the threshold of $0.05$, and therefore verifies the significance of our UDIL's performance improvement.
>
> For Permutation-MNIST, our UDIL's performance is very close to that of the best baseline, DER++ (92.666% versus 92.950%). Notably, the joint-training accuracy upper-bound, i.e., oracle, on this dataset only reaches an accuracy of 96.368%, *leaving little room for significant improvement*. Consequently, our model does not exhibit a dominantly superior performance on this dataset. We also did the hypothesis test for this dataset. The $p$-value of DER++ better than UDIL is $0.152$, which is much larger than the threshold $0.05$, therefore rejecting that the DER++ is significantly superior, and showing that UDIL is comparable to DER++ even on this dataset.
>
> Overall, the experiments demonstrate the efficacy of UDIL, particularly in scenarios where distributional shifts are more pronounced, highlighting its potential as a strong candidate for handling domain incremental learning challenges.
>
>
> **Q2. Details on the computational complexity/cost of the method are not provided in the formulation of the method or in the experiments. ...needs to learn the coefficients...Eq. 13 is more difficult than the optimization problem in counterpart methods, what would be the extra-cost in terms of computation?**
>
> Thank you for the suggestion. The overhead of optimizing the coefficients is *minimal* since it is a small Linear Programming (LP) problem, which involves only $3(t-1)$ variables where $t$ is the number of domains. For example, our Seq-CORe50 dataset has 11 domains, which leads to an LP problem with *only 30 variables*. Solving this LP problem usually takes less than a second; therefore the introduced overhead is negligible.
>
> It is worth noting that when optimizing (learning) the coefficients in Eqn. 13, the 0-1 loss does not involve any gradient calculation on the model $h$; all loss terms associated with the coefficients are treated as constants. This is why such optimization can be treated as a simple LP problem.
>
> We also provide a more detailed discussion on our UDIL's computation complexity in **Q1 of the General Response** above, and will include a similar discussion in the our revision as suggested.

---

> > ### Comment · Reviewer_4VUe · 2023-08-19
> > **Rebuttal from authors**
> >
> > Thanks to the authors for their responses to my questions. I consider this paper quite a strong contribution, and one of the best readings I've had as a NeurIPS reviewer in the past 5 years. I am very supportive of this paper being accepted.

---

> > > ### Author Response · Authors · 2023-08-19
> > > **Thank You**
> > >
> > > Thank you very much for the encouraging comments and for acknowledging our contribution. We are glad that our response has been helpful and convincing.

---

### Official Review · Reviewer_jN5u · 2023-07-05

**Soundness:** 3 good
**Presentation:** 3 good
**Contribution:** 3 good
**Rating:** 5
**Confidence:** 4

**Summary:**

In this paper, the author proposed a unified theoretical framework in order to reveal how the tasks in the domain incremental learning are related. Moreover, the framework unifies various existing methods from the perspective of domain generalization bound and it showed that the unified generalization bound can be organized in the format of the combination of data, memory bank, and history model. Meanwhile, the author showed that the fixed coefficient of the above combination is always less effective compared to the adaptive ones. As a result, the author proposed an adaptive adjusting coefficients method to tightly bound the generalization upper bound. Extensive experiments also showed the effectiveness of the unified framework.

**Strengths:**

1. This paper is well-organized and the mathematical proofs are sound. \\
2. The author combined theoretical analysis of domain adaptation to deal with the domain incremental problem, which is impressive. \\
3. The proposed unified framework can explain a variety of previous works, which is very powerful. \\

**Weaknesses:**

see questions

**Questions:**

1. The author should emphasize what the ghost example in supplementary material is ( if it is some kind of subset of the previous domain? or some stored replay examples ), otherwise the transition from Equation 19 to 20 is not smooth, e.g. it provided a loose scale-up. \\
2. From Theorem 3.4, we can find that the first bounding term is actually a
sum up of loss from the first model to the current one (ERM), so is that appropriate to store the model at every training stage in the sample replay setting, and is the history model froze during the training? Meanwhile, will that cause a lot of additional memory burden? \\
3. The intra-domain and cross-domain constraint is actually an every step alignment with the history learned models, thus as long as the domain discrepancy is relatively large from one to another, the $h$ would have a high probability of severe change, making the second term of Equation 8 in the left large, would that be a problem since it seems no explicit parameter regularization?
4. Also, since in Equation 8, every step of updates would come from a bottom-up calculation from the first domain to the current one, it seems the calculation would be expensive as the growth of the domain, so I have concerns over the calculation burden. Also, as a consequence, would the method can be implemented in some large-scale real-world datasets which used in domain incremental learning, e.g. DomainNet?
5. The Theorem 3.4 seems like a combination loss over all domains, thus, I have a little concern over the novelty of this Theorem. ( Considering it is the main Theorem).

**Limitations:**

 yes, the authors adequately addressed the limitations yet

---

> ### Author Rebuttal · Authors · 2023-08-07
>
> Thank you for the constructive and encouraging comments as well as the insightful questions. We are glad that you find our unified framework ``"powerful"`` and able to ``"explain a variety of previous works"``, our theory ``"sound"``/``"impressive"``, and our paper ``"well-organized"``. Below we address your questions one by one. **We will include the following discussions in our revision as suggested.**
>
> **Q1. ...what the ghost example in supplementary material is ...**
>
> The term "ghost sample" (or "ghost data") refers to a second dataset that is sampled from the same data distribution, but does not actually exist; rather, it serves solely as a tool for analysis. This is a *standard proving technique* [1, 2] for analyzing generalization bound. The reason for introducing ghost data is that measuring the true error $\hat{\epsilon}_{\alpha}(h)$ requires an infinite number of samples, making it challenging to apply the concept of VC-dimension for analysis. Therefore, in the inequality (Eq. 20), we follow the literature [1, 2] to introduce "ghost data" to facilitate analysis of the generalization bound.
>
> **Q2. ...is that appropriate to store the model at every training stage in the sample replay setting, and is the history model froze during the training? ...will that cause a lot of additional memory burden?**
>
> These are good questions. Actually we *only* need to store *one* model, i.e., the model trained after domain $t-1$. Note that the history model $H_{t-1}$ in Theorem 3.4 is *not* a collection of past models trained from the first domain to the most recent one, but rather the model trained after domain $t-1$, which has a constant memory consumption. As a result, when we calculate and optimize the first term in Theorem 3.4, we compute the discrepancy between the current model $h$ and the history model $H_{t-1}$ evaluated over different domains, requiring only one forward pass and backward pass. Therefore, memory cost is not a concern, making our UDIL scalable.
>
> Yes, the history model is frozen during the training.
>
> **Q3. ...as long as the domain discrepancy is relatively large from one to another, the $h$ would have a high probability of severe change, making the second term of Equation 8 in the left large, would that be a problem since it seems no explicit parameter regularization?**
>
> This is a good question. To avoid this problem, there is actually some implicit measure imposed by the *optimizable* term $\frac{1}{2}d\_{\mathcal{H}\Delta\mathcal{H}}(\mathcal{D}\_i, \mathcal{D}\_t)$ (note that this term is not a constant; instead it can be optimized in Eqn. 13) in Eqn. 7 (and Eqn. 8).
>
> **(1)** To see this, assuming $h$ consists of an encoder $e$ and a predictor $p$, minimizing this $d\_{\mathcal{H}\Delta\mathcal{H}}$ term (i.e., the third term of UDIL's objective function in Eqn. 8 or Eqn. 13) is equivalent to aligning previous domains $\mathcal{D}\_i$ and the current domain $\mathcal{D}\_t$ in the encoding space; this effectively regularizes the encoder $e$ of the model $h$ and minimizes the domain discrepancy in the encoding space, such that $h$ would not change severely.
>
> **(2)** Furthermore, our adaptively updated coefficient $\Omega$ is inherently designed for addressing this issue: treating Eqn. 8 as the objective function to minimize, if the discrepancy between two domains $d\_{\mathcal{H}\Delta\mathcal{H}}(\mathcal{D}\_i, \mathcal{D}\_t)$  is too large for the adversarial training to align their features, the corresponding coefficient $\beta_i$ in the third term of Eqn. 8, $\beta\_i \cdot d\_{\mathcal{H}\Delta\mathcal{H}}(\mathcal{D}\_{i}, \mathcal{D}\_{t})$, will be adapted to a very small value. Since $\beta\_i$ also controls the second term of Eqn. 8, $\beta\_i \cdot \hat{\epsilon}_{\mathcal{D}\_t}(h, H\_{t-1})$, smaller $\beta\_i$ will "de-emphasize" the cross-domain loss term $\hat{\epsilon}\_{\mathcal{D}\_t}(h, H\_{t-1})$, thereby minimizing the negative effect of optimizing this term.
>
> **Q4. ...calculation...expensive as the growth of the domain...large-scale realistic dataset...DomainNet...**
>
> Please refer to **Q1 & Q2 of the General Response** above.
>
> **Q5. ...Theorem 3.4 seems like a combination loss over all domains...the novelty of this Theorem.**
>
> We are sorry for the confusion. We should have better highlighted our novelty in the main paper. While Theorem 3.4 presents a combination of losses over all domains, it has novelty that can be outlined in three aspects:
>
> (i) Theorem 3.4 introduces the first unified theoretical framework that provides a generalization bound encompassing various existing practical techniques (intra-domain and cross-domain distillation loss) in Domain Incremental Learning (DIL). This comprehensive approach, as illustrated in Table 1 of the paper, sets it apart from prior work.
>
> (ii) Another key innovation of Theorem 3.4 is the provision for automatically learning adaptive coefficients for different loss terms. This capability plays a pivotal role in achieving the tightest generalization bound and ultimately enhancing the model's performance.
>
> (iii) Theorem 3.4 systematically decomposes the joint-training objective of the general DIL problem into an Empirical Risk Minimization (ERM) term, an intra-domain term, and a cross-domain term, taking into account the domain discrepancy. This offers valuable insights for more inclusive (encompassing more terms into the bound) and stronger (developing a tighter bound) frameworks in the future.
>
> Reviewer 4VUe also commends the novelty of our UDIL, acknowledging it as a ``"significant"`` contribution and ``"the first comprehensive analysis of generalization bounds in domain incremental learning while also categorizing SOTA methods in the area according to different views of these bounds"``.
>
> **References**
>
> [1] Abu-Mostafa, et al. *Learning from data*. AMLBook, 2012.
>
> [2] Mohri, et al. *Foundations of machine learning*. MIT press, 2018.

---

### Official Review · Reviewer_Ps1S · 2023-07-15

**Soundness:** 3 good
**Presentation:** 3 good
**Contribution:** 3 good
**Rating:** 6
**Confidence:** 3

**Summary:**

This submission investigates domain incremental learning problem, where the model is adapted to a sequence of new tasks which have different distributions. Starting from some (domain related) generalization bound theory, the combination of three different bounds could be used to address domain incremental learning, and meanwhile several existing methods could be unified in the combined bound. An objective to find the efficient combination weight is also introduced. Experiments on both toy dataset and real dataset show the efficacy of the method.

**Strengths:**

- The motivation is clear, it is reasonable to combine the three different bounds to deal with mode general case.

- It is good to derive a decent method from theoretic perspective for domain incremental learning task. And it is creditable that some hyperparameters such as $C$ could be also estimated inside the bound.

- It is admirable that the resulting bound could include several existing methods, which connects the intuitive methods and the corresponding theory.



**Weaknesses:**

Overall the submission is a good paper, and I have no obvious concerns about the submission, while I simply hope maybe the authors could add more narrations in the main paper, on the related works of domain incremental learning, as there are relatively fewer existing works in this topic. And I am also interested in the effectiveness on the huge dataset, such as DomainNet which has relatively larger domain gap and more categories.

**Questions:**

As the generalization bounds considered in the submission are actually related to domain adaptive task (from [3]), I am curious about whether the method could be extended to the more practical scenario, unsupervised domain incremental learning, where the new domain only contains unlabeled samples. I think there is possibility to extend the work, as some unsupervised domain adaptation methods are directly utilizing the $d_{\mathcal{H}\Delta\mathcal{H}}$, such as MCD[a], though in the current setting there are only limited source samples.



*reference*

[a] Saito K, Watanabe K, Ushiku Y, et al. Maximum classifier discrepancy for unsupervised domain adaptation[C]//Proceedings of the IEEE conference on computer vision and pattern recognition. 2018: 3723-3732.

**Limitations:**

None potential negative societal impact.

---

> ### Author Rebuttal · Authors · 2023-08-07
>
> Thank you for the constructive and encouraging comments as well as the insightful questions. We are glad that you find our motivation ``"clear"``/``"reasonable"``,  our method ``"decent"``/``"creditable"``, our theoretical analysis ``"admirable"``, and our submission ``"a good paper"`` with ``"no obvious concerns"``. Below we address your questions one by one.
>
> **Q1. Maybe the authors could add more narrations in the main paper about the related work of DIL?**
>
> This is a good suggestion. In the revised version of our paper, we will follow your suggestion to include a more detailed introduction to the field of Domain-Incremental Learning (DIL). We observe that while DIL is one of the three main categories in continual learning, there is a lack of systematic study specifically focused on this topic, in comparison to Task-Incremental Learning (TIL) and Class-Incremental Learning (CIL). We hope our work could contribute to the understanding and development of DIL methods and address the challenges in this exciting but underexplored area.
>
> **Q2. I am curious about whether the method could be extended to the more practical scenario, unsupervised domain incremental learning?**
>
> Thanks for this insightful question. Indeed it would be very interesting to extend our UDIL framework to unsupervised domain incremental learning. We expect that the availability of the next domain's (unlabeled) data would offer more legitimacy for aligning domains' feature distributions (similar to MCD (cited as [44] in our paper) in the unsupervised domain adaptation setting). For now, our work primarily focuses on the standard DIL setting with memory, providing a unified perspective that incorporates previously existing DIL techniques. We hope that this contribution will lay a solid foundation for future research, including the exploration of the unsupervised domain incremental learning idea you proposed.
>
>
> **Q3. I am also interested in the effectiveness on the huge dataset, such as DomainNet which has relatively larger domain gap and more categories.**
>
> Please refer to **Q2 of the General Response** above.

---

> > ### Comment · Reviewer_Ps1S · 2023-08-16
> >
> > Thanks for the comments, I will keep my original scores.

---

### Author Rebuttal · Authors · 2023-08-07

# General Response

We thank all the reviewers for their valuable and constructive comments. We are glad they find the problem we address ``"original"``/``"of high relevance"`` (4VUe, Ps1S), our proposed unified framework ``"admirable"``/``"reasonable"``/``"powerful"``/``"original"`` (Ps1S, jN5u, 4VUe), our theoretical analysis ``"impressive"``/``"sound"``/``"rigorous"`` (Ps1S, jN5u, 4VUe), our paper ``"clear"``/``"well-organized"``/``"well written"`` (Ps1S, jN5u, 4VUe), and acknowledge that our experiments show the ``"efficacy"``/``"effectiveness"`` of our method (Ps1S, jN5u) and that our UDIL is ``"the first comprehensive analysis of generalization bounds in domain incremental learning"``. Below we address reviewers' two common questions.

**Q1. \[UDIL's Computational Complexity?\]** *"...in Equation 8, every step of updates would come from a bottom-up calculation from the first domain to the current one, it seems the calculation would be expensive as the growth of the domain...the calculation burden..."* **(jN5u)**; *"Details on the computational complexity/cost of the method are not provided in the formulation of the method or in the experiments. ...Eq. 13 is more difficult than the optimization problem in counterpart methods, what would be the extra-cost in terms of computation?"* **(4VUe)**

This is a good question. In fact, each step of update does not grow with the number of domains $t$, which makes our algorithm scalable.

Note that in general, UDIL's memory bank has a **constant** number of data points; as the number of domain increases, the number of data points for each domain decreases (refer to Appendix C.2).
Therefore, while one needs to calculate the loss from the first domain to the current one, such calculation takes constant time and **does not grow with the number of domains**.

Below, we provide a more detailed analysis of UDIL's computational complexity. As shown in Eqn. 13, our framework calculates three loss functions (stopping-gradient operation omitted for clarity):

$
V_l(h, \Omega) + V_{\text{0-1}}(h, \Omega) + V_d(d, e, \Omega).
$

The computational cost of optimization of the first term $V_l$ is the same as our replay-based baselines such as vanilla experience replay (ER) and dark experience replay (DER++); this term estimates and optimizes the classification loss with respect to the model parameters.

Therefore, UDIL's computational overhead comes from the last two terms, $V_{\text{0-1}}(h, \Omega)$ and $V_d(d, e, \Omega)$:

**(1)** $V_{\text{0-1}}(h, \Omega)$ optimizes the coefficients to find the tightest upper bound for UDIL. Its 0-1 loss does not involve any gradient calculation on the model $h$, i.e., all loss terms associated with the coefficients are treated as constants; therefore the **overhead of optimizing the coefficients is minimal** since it is a **small Linear Programming problem** (only $3(t-1)$ variables where $t$ is the number of domains). For example, our Seq-CORe50 dataset has 11 domains, which leads to an LP problem with *only 30 variables*. Solving this LP problem usually takes *less than a second*; therefore the introduced overhead is *negligible*.

**(2)** $V_d(d, e, \Omega)$ involves training a small multiclass domain classifier $d$ on top of the encoder $e$. Its computational cost is almost the same as $V_l(h, \Omega)$. Consequently, the computational complexity of our proposed framework *remains in the same order as the baselines such as ER and DER++*.

The ``"bottom-up calculation from the first domain to the current one"``, as mentioned by Reviewer jN5u, is achieved by concatenating the mini-batches of data from different domain, and can be effectively calculated without any additiional overhead.

In summary, we can see that our UDIL's computational complexity is in the same order as the strongest baselines such as ER and DER++; therefore UDIL is scalable to large datasets.

**Q2. \[UDIL's Application to Large-Scale Datasets?\]** *"I am also interested in the effectiveness on the huge dataset, such as DomainNet which has relatively larger domain gap and more categories"* **(Ps1S)**; *"would the method can be implemented in some large-scale real-world datasets which used in domain incremental learning, e.g. DomainNet?"* **(jN5u)**.

Following your suggestion, we ran additional experiments on the DomainNet dataset. Considering the constraints of time and computational resources, we compared our UDIL with only DER++ in these additional experiments, as DER++ is our strongest baseline across four datasets. We also report the performance of the fine-tuning baseline and joint-training oracle. Due to the constraint of computational resources, we select the ResNet-18 as our backbone network. We grid-searched the learning rate in the range of (1e-4, 5e-4, 1e-3, 5e-3, 1e-2) with Adam optimizer for all four methods, and we inherit the rest of hyper-parameters of UDIL from the Seq-CORe50 dataset. The following table shows the average accuracy (Avg. Acc.) and forgetting rates of UDIL and two baseline models Fine-Tune and DER++, as well as the oracle model Joint-Training.

| **Model** | **Avg. Acc. (↑)** | **Forgetting (↓)** |
|:-----------:|:--------------------:|:---------------:|
| Fine-Tune | 19.675 | 24.971 |
| DER++ | 22.996 | 17.043 |
| UDIL (Ours) | **24.232** | **14.171** |
| Joint-Training (Oracle) | 38.582 | - |
| | |

As shown in the table, there is a noticeable and significant performance boost (considering DomainNet is a 347-way classification problem) when applying UDIL, which further validates the effectiveness of our proposed methods on the large-scale realistic dataset. We will conduct more systematic experiments on the DomainNet dataset in the revised version of our work, including a more thorough search over the hyper-parameters and experiments on other baseline models.

---

### Decision · Program_Chairs · 2023-09-21

**Decision:**

Accept (poster)

**Comment:**

The paper has been positively evaluated by three reviewers (5,6,7 scores). The provided rebuttal and discussion did not result in further changes in the score. The reviewers agree that the paper offers new insights on domain incremental learning, and that the theoretical analysis is of interest to the community. The AC recommends acceptance of the paper. The authors should incorporate the additional results and discussion on computational complexity in the final version.